# Intrinsic mechanisms in the gating of resurgent Na+ currents

**Joseph L Ransdell[1][†][‡], Jonathan D Moreno[2][†], Druv Bhagavan[2], Jonathan R Silva[2], Jeanne M Nerbonne[1,3]\***

[1]Departments of Medicine, Cardiovascular Division, Washington University, St. Louis, United States; [2]Developmental Biomedical Engineering, Washington University, St. Louis, United States; [3]Developmental Biology, Washington University, St. Louis, United States

**Abstract** The resurgent component of the voltage-gated sodium current ($I_{NaR}$) is a depolarizing conductance, revealed on membrane hyperpolarizations following brief depolarizing voltage steps, which has been shown to contribute to regulating the firing properties of numerous neuronal cell types throughout the central and peripheral nervous systems. Although mediated by the same voltage-gated sodium (Nav) channels that underlie the transient and persistent Nav current components, the gating mechanisms that contribute to the generation of $I_{NaR}$ remain unclear. Here, we characterized Nav currents in mouse cerebellar Purkinje neurons, and used tailored voltage-clamp protocols to define how the voltage and the duration of the initial membrane depolarization affect the amplitudes and kinetics of $I_{NaR}$. Using the acquired voltage-clamp data, we developed a novel Markov kinetic state model with parallel (fast and slow) inactivation pathways and, we show that this model reproduces the properties of the resurgent, as well as the transient and persistent, Nav currents recorded in (mouse) cerebellar Purkinje neurons. Based on the acquired experimental data and the simulations, we propose that resurgent Na+ influx occurs as a result of fast inactivating Nav channels transitioning into an open/conducting state on membrane hyperpolarization, and that the decay of $I_{NaR}$ reflects the slow accumulation of recovered/opened Nav channels into a second, alternative and more slowly populated, inactivated state. Additional simulations reveal that extrinsic factors that affect the kinetics of fast or slow Nav channel inactivation and/or impact the relative distribution of Nav channels in the fast- and slow-inactivated states, such as the accessory Navβ4 channel subunit, can modulate the amplitude of $I_{NaR}$.

## Editor's evaluation

After more than 20 years of intensive research the molecular machinery of Resurgent Currents (INaR), a non-canonical identity of currents mediated by voltage-activated sodium channels is still a mystery. In this paper, Ransdell and colleagues advance the conceptual framework with new experimental insight and a new kinetic model of INaR.

## Introduction

Voltage-gated sodium (Nav) channels open rapidly on membrane depolarization and underlie the generation of action potentials in many excitable cells, including skeletal and cardiac muscle, as well as central and peripheral neurons. The pore-forming (α) subunits of Nav channels, Nav1.1 to Nav1.9, belong to the 'S4' superfamily of voltage-gated ion channel genes (*Catterall, 2010*). Each Nav α subunit comprises four homologous domains (DI–DIV) with six transmembrane spanning segments (S1–S6) (*Noda et al., 1984*). The S1–S4 segments in each domain form the four voltage sensing

**\*For correspondence:**
jnerbonne@wustl.edu

[†]These authors contributed equally to this work

**Present address:** [‡]Miami University, Oxford, OH, United States

**Competing interest:** The authors declare that no competing interests exist.

domains (VSDs) that activate on membrane depolarization (*Noda et al., 1984*; *Guy and Seetharamulu, 1986*), resulting in channel opening and Na$^+$ influx. Nav channels conduct Na$^+$ when the VSDs of domains I, II, and III move outwardly to an activated conformation (*Bezanilla, 2000*). Following opening, fast inactivation occurs (*Bezanilla and Armstrong, 1977*), mediated by a hydrophobic (IFM) motif in the cytosolic DIII–DIV linker that binds to a site near the pore that is revealed on activation of the DIII and DIV VSDs (*Bosmans et al., 2008*; *Capes et al., 2013*; *Hsu et al., 2017*; *Cha et al., 1999*). Thus, fast inactivation of open Nav channels occurs when all four VSDs are in the outward/activated position (*Capes et al., 2013*). If domains I, II, and III are activated and DIV is in the deactivated position, however, the IFM motif does not bind, which results in the generation of a non-inactivating or persistent Nav current ($I_{NaP}$) component (*Chanda and Bezanilla, 2002*; *Horn et al., 2000*).

An additional Nav current component, that is observed on membrane *hyperpolarizations* following brief depolarizing voltage steps and referred to as the resurgent component of Nav current ($I_{NaR}$), was first described in isolated, postnatal day 8–14 rat cerebellar Purkinje neurons (*Raman and Bean, 1997*). Although linked to the regulation of the spontaneous firing of action potentials in cerebellar neurons (*Raman and Bean, 1997*; *Raman and Bean, 1999*), $I_{NaR}$ was subsequently identified in more than 20 types of neurons in the central and peripheral nervous systems, only some of which are spontaneously active (*Lewis and Raman, 2014*) suggesting that $I_{NaR}$ likely plays diverse functional roles in regulating neuronal excitability. Although flowing through the same Nav channels as the transient ($I_{NaT}$) and persistent ($I_{NaP}$) sodium current components, the time- and voltage-dependent properties of $I_{NaR}$ are distinct (*Lewis and Raman, 2014*). In addition to being revealed on membrane *hyperpolarizations* presented after brief (~5 ms) depolarizing voltage steps that evoke $I_{NaT}$ (*Khaliq et al., 2003*), for example, the time courses of $I_{NaR}$ activation and decay are much slower than $I_{NaT}$. These experimental observations were interpreted as suggesting a Nav channel gating model with two distinct mechanisms contributing to inactivation: a conventional, fast inactivation mechanism in which channels recover from inactivation without passing through an open conducting state; and, a second mechanism, favored by brief depolarizations, in which channels recover from inactivation by passing through an open, conducting state (*Raman and Bean, 2001*). It was further suggested that the second mechanism was consistent with a voltage-dependent process whereby Nav channels, opened on depolarization, are blocked by an endogenous 'blocking' particle that occludes the pore, driving channels into an 'open-blocked' (OB) state (*Raman and Bean, 2001*). On subsequent membrane hyperpolarization, the blocking particle is displaced, and Na$^+$ flows through unblocked/open Nav channels, generating the resurgent Nav current (*Raman and Bean, 2001*). Importantly, in this gating scheme, blocked Nav channels do not inactivate and inactivated channels are not blocked, that is, a Nav channel cannot be blocked and inactivated simultaneously (*Lewis and Raman, 2014*).

Studies focused on defining the molecular mechanism(s) underlying the generation of $I_{NaR}$ revealed that proteases (e.g., trypsin/chymotrypsin) that act at positively charged and aromatic/hydrophobic amino acid residues eliminate $I_{NaR}$, while increasing $I_{NaT}$, observations interpreted as supporting the blocking particle model and suggesting that the putative blocker was a protein within the Nav channel complex (*Grieco et al., 2002*). Attention focused quickly on the transmembrane accessory Navβ4 subunit, which has a short cytosolic tail with several positive charges and multiple aromatic/hydrophobic residues (*Grieco et al., 2005*). Clear support for a role for Navβ4 was provided in experiments in which intracellular application of a synthetic Navβ4 peptide containing the tail sequence (β4154–167), following elimination of the resurgent Nav current by trypsin or chymotrypsin, rescued $I_{NaR}$ in isolated cerebellar Purkinje neurons (*Grieco et al., 2005*). In addition, experiments on CA3 pyramidal neurons, which lack $I_{NaR}$ and Navβ4, demonstrated that the application of the β4154–167 peptide generated resurgent Nav currents (*Grieco et al., 2005*). Further support for a critical role for Navβ4 was provided with the demonstration that treatment of mouse cerebellar granule neurons with small interfering RNAs (siRNAs) targeting *Scn4b* (Navβ4) resulted in the loss of $I_{NaR}$ and that the subsequent exposure of *Scn4b*-siRNA-treated cells to the β4154–167 peptide rescued $I_{NaR}$ (*Grieco et al., 2005*; *Bant and Raman, 2010*).

In experiments designed to test directly the hypothesis that Navβ4 is *required* for the generation of $I_{NaR}$, however, we found that $I_{NaR}$ was reduced (by ~50%), but *not* eliminated, and that the time- and voltage-dependent properties of the currents were unaffected in cerebellar Purkinje neurons in (*Scn4b$^{-/-}$*) mice lacking Navβ4 (*Ransdell et al., 2017*). It was subsequently reported that $I_{NaR}$ densities in cerebellar Purkinje neurons isolated from another *Scn4b$^{-/-}$* mouse line were not significantly different

from the currents measured in wild type cells (*White et al., 2019*). In addition, it has been reported that $I_{NaR}$ is readily measured in *Scn4b$^{-/-}$* striatal neurons (*Miyazaki et al., 2014*). These observations clearly suggest that additional mechanisms contribute to the generation of $I_{NaR}$. One possibility is that there are other open channel blocking molecules expressed in Purkinje (and other) neurons. Support for this hypothesis was provided in studies showing that $I_{NaR}$ is decreased in neonatal mouse cerebellar Purkinje neurons isolated from animals harboring a targeted disruption in the *Fgf14* (which encodes intracellular fibroblast growth factor 14 [iFGF14]) locus (*White et al., 2019*), as well as in wild type neonatal Purkinje cells following exposure to an interfering RNA targeting the *Fgf14b* variant (*Yan et al., 2014*). Interestingly, however, it was also reported that $I_{NaR}$ was readily detected in Purkinje neurons isolated from neonatal animals lacking both *Scn4b* and *Fgf14* (*White et al., 2019*). It is certainly possible and that there are additional, yet to be discovered, endogenous open channel blockers that contribute to the generation of $I_{NaR}$. Alternatively, it seemed possible to us that there is an intrinsic gating mechanism(s) by which Nav channels can produce resurgent current. Here, we present the results of experimental and modeling efforts designed to explore the latter hypothesis

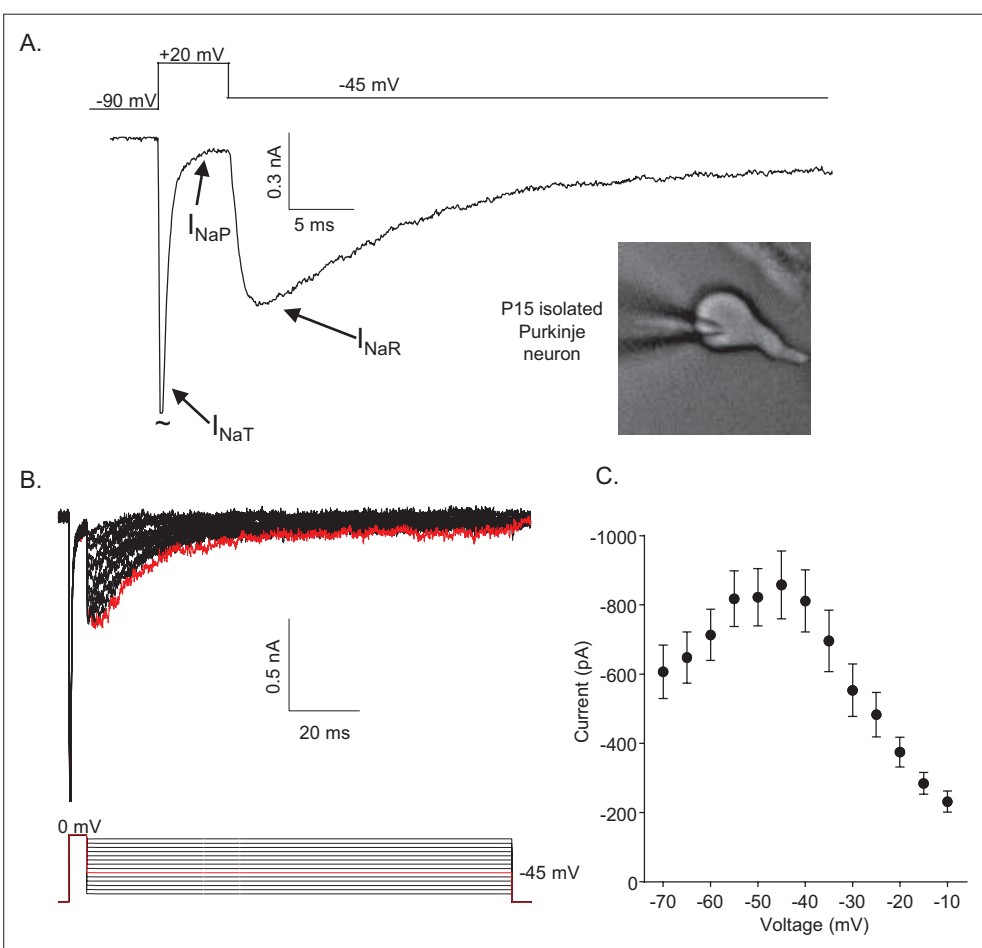

**Figure 1.** Mouse cerebellar Purkinje neurons express three voltage-gated sodium (Nav) current components. (**A**) Representative recording of the transient ($I_{NaT}$), persistent ($I_{NaP}$), and resurgent ($I_{NaR}$) components of the Nav currents in an isolated neonatal mouse cerebellar Purkinje neuron. The voltage-clamp paradigm is displayed above the current record, and the labelled arrows indicate the three Nav current components. (**B**) $I_{NaR}$ waveforms, recorded during hyperpolarizing voltage steps to various potentials ranging from –70 to –10 mV, following 5 ms depolarizing voltage steps to 0 mV from a holding potential (HP) of –80 mV; the voltage-clamp paradigm is shown below the current records. The current record highlighted in red was recorded during the –45 mV hyperpolarizing voltage step (also indicated in *red* in the illustrated voltage-clamp paradigm). (**C**) Mean ± SEM (n = 15) peak $I_{NaR}$ amplitudes are plotted as a function of the hyperpolarizing test potential; the peak $I_{NaR}$ is recorded at approximately –45 mV.

directly, and we provide evidence for a 'blocking-particle independent' mechanism in the generation of $I_{NaR}$ in cerebellar Purkinje neurons.

## Results

### The amplitude of $I_{NaR}$ depends on the duration, but not the voltage, of the prior membrane depolarization

In isolated neonatal (P12–P16) mouse cerebellar Purkinje neurons, the fast transient ($I_{NaT}$), persistent ($I_{NaP}$), and resurgent ($I_{NaR}$) Nav current components can be distinguished using voltage-clamp protocols that take advantage of the unique time- and voltage-dependent properties of the three current components (*Figure 1A*). On membrane depolarization, for example, $I_{NaT}$ activates fast and subsequently decays rapidly to a steady-state (persistent) level of inward current, $I_{NaP}$ (*Figure 1A*). $I_{NaR}$, in contrast, is revealed on membrane *hyperpolarizations* from the depolarized membrane potentials that evoke $I_{NaT}$ (*Figure 1A*). In addition, the time courses of $I_{NaR}$ activation and decay are much slower than $I_{NaT}$ activation and decay (*Figure 1A*). Additional experiments revealed that, in response to membrane hyperpolarizations following brief (5 ms) depolarizing steps (to 0 mV), the amplitude of $I_{NaR}$ varies as a function of the membrane potential of the hyperpolarizing voltage step (*Figure 1B*). The maximal amplitude of $I_{NaR}$ is observed at approximately –45 mV (*Figure 1C*).

To determine how the duration and the voltage of the depolarizing voltage step (that evokes $I_{NaT}$) affect the amplitudes and waveforms of $I_{NaR}$, voltage-clamp paradigms were designed in which either the duration or the voltage of the depolarizing step was varied (*Figure 2*). In initial experiments, the external and internal $Na^+$ concentrations were 151 and 8 mM, respectively, resulting in a $Na^+$ reversal potential of +75 mV. These experiments revealed that prolonging the duration of the +20 mV depolarizing voltage step resulted in the marked attenuation of the peak amplitudes of $I_{NaR}$ measured during hyperpolarizing voltage steps to –45 mV (*Figure 2A*). The time course of the attenuation of $I_{NaR}$ was well described by a single exponential characterized by a mean ± SEM (n = 12) time constant of 15.5 ± 0.5 ms (*Figure 2B*).

Subsequent experiments explored the effect of the driving force on $Na^+$ on the time-dependent attenuation of $I_{NaR}$. In these experiments, the extracellular $Na^+$ was reduced to 50 mM and the $Na^+$ concentration in the internal solution was increased to 15 mM, resulting in a $Na^+$ reversal potential of approximately +30 mV. Under these recording conditions, depolarizations to +46 mV resulted in outward $I_{NaT}$ (*Figure 2C*). Similar to the results obtained with inward $I_{NaT}$ (*Figure 2A and B*), prolonging the depolarizing (+46 mV) voltage step when $I_{NaT}$ is outward results in the rapid attenuation of the amplitudes of $I_{NaR}$ evoked during the subsequent hyperpolarizations to –45 mV (*Figure 2C*). Under these conditions (outward $I_{NaT}$), the time course of the attenuation of $I_{NaR}$ was also well described by a single exponential with a mean ± SEM (n = 6) time constant of 13.8 ± 1.1 ms (*Figure 2C*), a value that is very similar to that observed when $I_{NaT}$ is inward (*Figure 2B*). Taken together, these combined results demonstrate that, following brief depolarizations, there is a time-dependent accumulation of Nav channels (which underlie $I_{NaR}$) in a non-conducting state, and that this accumulation occurs independent of the direction of the movement of permeating $Na^+$ ions.

To determine how the voltage of the depolarizing step affects the amplitudes and kinetics of $I_{NaR}$, the currents recorded at –45 mV after 5 ms depolarizing steps to various membrane potentials (*Figure 2D*) were measured. These experiments revealed that hyperpolarizations to –45 mV following brief (5 ms) depolarizations to various membrane potentials (ranging from –45 to +10 mV) resulted in identical $I_{NaR}$ amplitudes (*Figure 2E*). Additionally, varying the voltage of the depolarizing step did not affect the kinetics of the decay of $I_{NaR}$ (*Figure 2D*). This is clearly illustrated in *Figure 2F*, in which Nav currents recorded (in the same cell) at –40 mV during a sustained voltage step (*red*) and following a 5 ms depolarizing voltage step to +10 mV are superimposed. The opening of Nav channels that conduct $I_{NaR}$, therefore, is not affected by the voltage of the prior membrane depolarization. Taken together, these observations suggest that there are two parallel, and kinetically distinct, Nav channel inactivation pathways: a fast inactivating pathway that is responsible for $I_{NaT}$; and, a second, slower inactivation pathway that underlies $I_{NaR}$.

### A novel Markov model, with parallel inactivation pathways, for Nav channel gating in Purkinje neurons

The results of the voltage-clamp experiments described above suggest that there are (at least) two distinct inactivation pathways that contribute to the gating of the Nav channels expressed in mouse

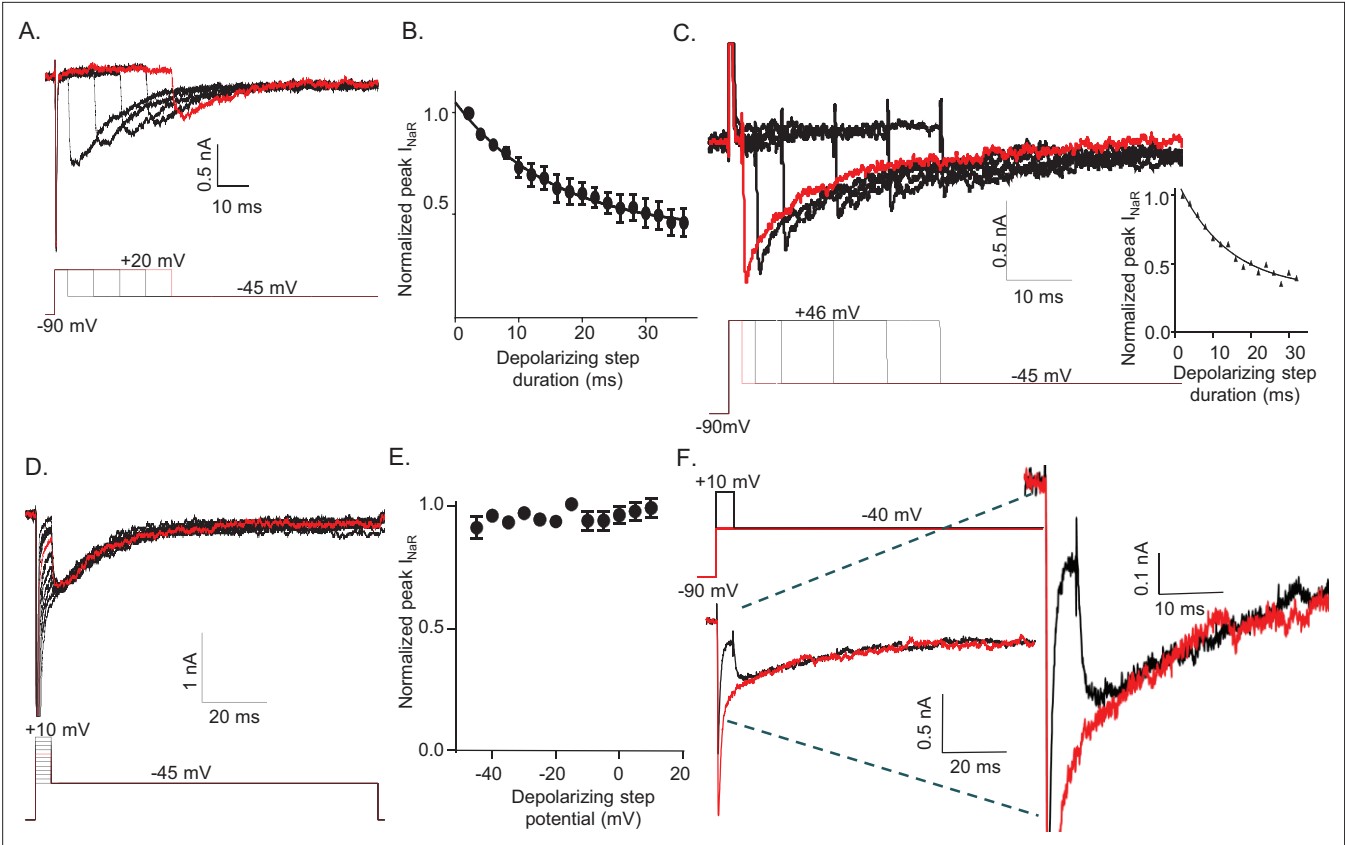

**Figure 2.** The amplitude of resurgent voltage-gated sodium current ($I_{NaR}$) is determined by the duration of the prior membrane depolarization. (**A**) In a neonatal mouse cerebellar Purkinje neuron, $I_{NaR}$ was revealed on membrane hyperpolarizations following depolarizing voltage steps to +20 mV of varying durations; the voltage-clamp paradigm is shown below the current records. (**B**) Peak $I_{NaR}$ amplitudes, evoked at –45 mV following each depolarizing voltage step to +20 mV, were measured and normalized to the maximal peak $I_{NaR}$ (measured in the same cell). The mean ± SEM (n = 12) normalized peak $I_{NaR}$ amplitudes are plotted as a function of the duration of the depolarizing voltage step. The attenuation of peak $I_{NaR}$ as a function of the duration of the depolarizing voltage step was well described by a single exponential with a mean ± SEM time constant of 15.5 ± 0.5 ms (n = 12). (**C**) The dependence of $I_{NaR}$ on the duration of the depolarizing voltage step was also measured with reduced (50 mM) extracellular and increased intracellular (15 mM) sodium, resulting in a $Na^+$ reversal potential of +30 mV. Under these conditions, depolarizing voltage steps to +46 mV evoked outward $I_{NaT}$. Peak $I_{NaR}$ amplitudes, revealed during hyperpolarizing voltage steps to –45 mV, however, were also found to vary as a function of the duration of the depolarizing voltage step, revealing that $I_{NaR}$ and the time-dependent attenuation of $I_{NaR}$ are not affected by the direction (inward versus outward) of $Na^+$ flux during the depolarizing voltage step. The peak amplitudes of $I_{NaR}$, evoked at –45 mV following each depolarizing voltage step, were measured in each cell and normalized to the maximal $I_{NaR}$ amplitude (in the same cell). As is evident from the representative records and the plot of normalized peak $I_{NaR}$ amplitudes (on the right), the attenuation of $I_{NaR}$ as a function of the duration of the depolarizing voltage steps to +46 mV is also well described by a single exponential with a mean ± SEM time constant of 13.8 ± 1.1 ms (n = 6), a value similar to that observed when $I_{NaT}$ was inward (**B**). (**D**) Representative $I_{NaR}$ waveforms, recorded directly on repolarizations to –45 mv following 5 ms depolarizations to various membrane potentials from a –90 mV HP, are shown; the voltage-clamp protocol is shown below the current records. (**E**) The mean ± SEM (n = 6) normalized peak $I_{NaR}$ amplitudes are plotted as a function of potential of the depolarizing voltage step. (**F**) Representative voltage-clamp recordings of Nav currents evoked (in the same cell) on direct depolarization to –40 mV from an HP of –90 mV (*red*) and on hyperpolarization to –40 mV following a 5 ms depolarizing voltage step to +10 mV (*black*) from the same HP; the voltage-clamp protocols are shown above the current records and the currents are shown on an expanded scale on the right. In panels A, C, D, and F, the currents in red were recorded during the voltage-clamp paradigms (shown below or above) depicted in red.

cerebellar Purkinje neurons, that is, one that is populated quickly on channel opening and inactivates rapidly (fast inactivation), and a second that is populated and decays much more slowly (slow inactivation). To explore this hypothesis, we developed a Markov kinetic state model that, after numerical optimization (see Materials and methods), recapitulates the range of time- and voltage-dependent properties observed experimentally for the Nav currents in mouse cerebellar Purkinje neurons. The optimized Markov model (**Figure 3A**) includes parallel fast (IF1, IF2) and slow inactivation (IS) pathways to reconcile the experimental findings (**Figure 2**) that the duration of the depolarizing voltage step that underlie Nav channel activation, and not the potential of the depolarizing voltage step or the

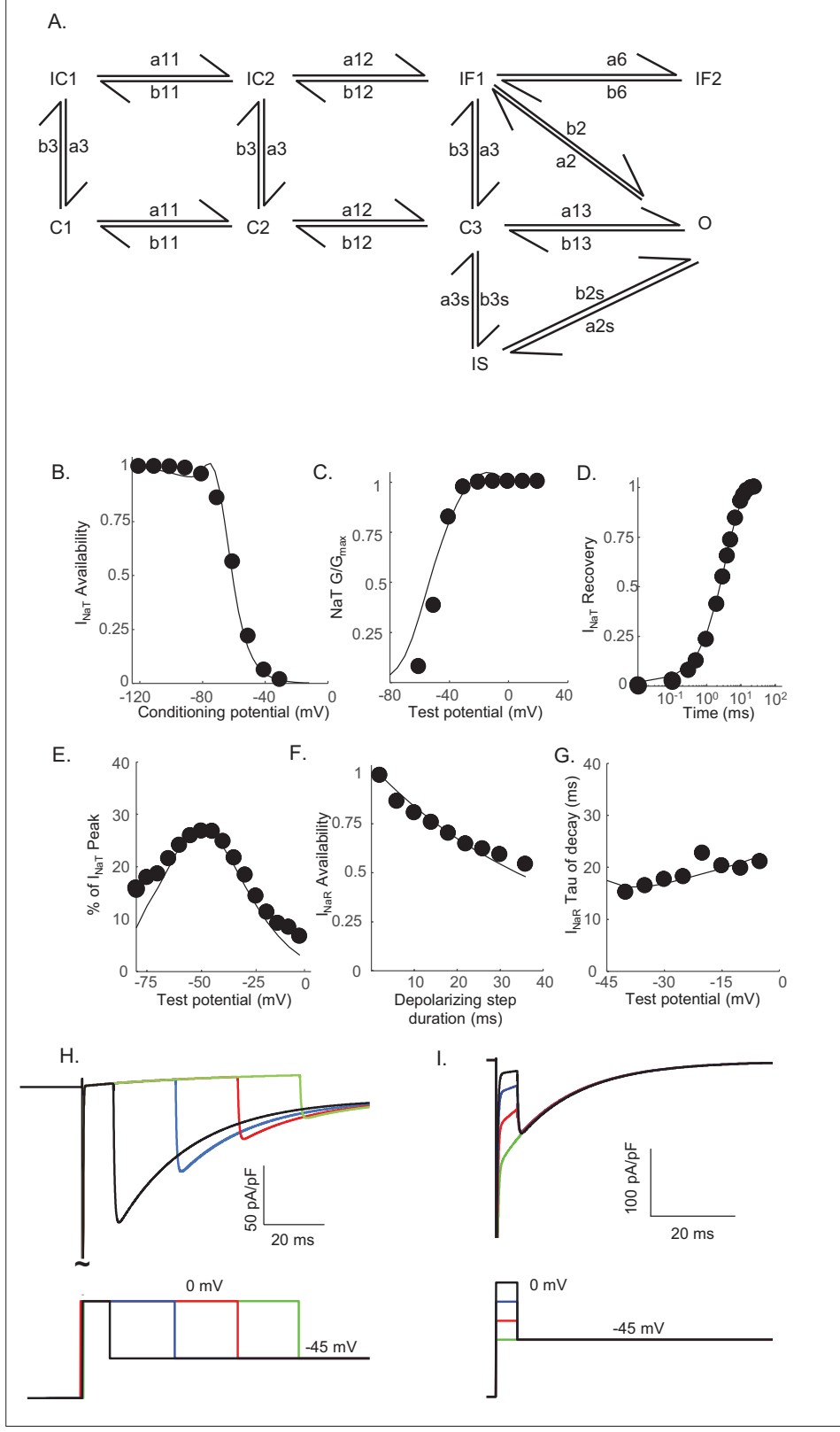

**Figure 3.** Novel Markov kinetic state model of voltage-gated sodium (Nav) channel gating in cerebellar Purkinje neurons. A novel Markov kinetic state model was developed with parallel fast inactivating (IF1, IF2) and slow inactivating (IS) gating pathways (**A**). The model was numerically optimized (see Materials and methods) by simulating the data generated using voltage-clamp protocols identical to those used in the experiments to

*Figure 3 continued*

determine the detailed time- and voltage-dependent properties of the Nav currents in cerebellar Purkinje neurons. The various rate constants in the model were numerically optimized to recapitulate the measured properties of $I_{NaT}$, $I_{NaP}$, and $I_{NaR}$ including the voltage dependences of steady-state inactivation (**B**) and activation (**C**) of $I_{NaT}$, and the kinetics of $I_{NaT}$ recovery from inactivation (**D**). The model also reproduces the measured properties of $I_{NaR}$, including the magnitude of $I_{NaR}$ relative to $I_{NaT}$ (**E**), the attenuation of the peak $I_{NaR}$ amplitude as a function of the duration of the depolarizing voltage steps (**F**), and the kinetics of the decay (inactivation) of peak $I_{NaR}$ amplitudes (**G**). Filled circles represent the mean experimental data and the lines represent the results of the simulation. The model successfully reproduces the observed, time-dependent attenuation of peak $I_{NaR}$ amplitudes that is evident experimentally on membrane hyperpolarizations following depolarizing voltage steps of varying durations (**H**), and the finding that the peak amplitude of $I_{NaR}$ is not affected by the potential of the depolarizing voltage step (**I**).

direction (i.e., inward or outward) of the Na$^+$ flux through open channels, determines the amplitudes of $I_{NaR}$ recorded during subsequent membrane hyperpolarizations. The model was constrained to fit the experimental data derived from multiple voltage-clamp protocols designed to detail the properties of $I_{NaT}$, including those to determine the voltage dependence of $I_{NaT}$ activation and steady-state inactivation, the time course of $I_{NaT}$ recovery from inactivation (*Figure 3B–D*), and the time constant (tau) of decay of the peak $I_{NaT}$. The model was further constrained by the experimental data obtained using protocols designed to detail the properties of $I_{NaR}$, including the voltage dependence of the ratio of the amplitudes of $I_{NaR}$ and $I_{NaT}$ ($I_{NaR}$:$I_{NaR}$), the time-dependent attenuation of $I_{NaR}$ amplitudes, observed as a function of the duration of the depolarizing voltage step (*Figure 2A and B*), and the time constants (tau) of $I_{NaR}$ decay determined for the currents recorded during hyperpolarizing voltage steps to various membrane potentials (*Figure 3E–G*). Consistent with the experimental results presented in *Figure 2*, simulations using this gating model reveal that the amplitude of $I_{NaR}$ is dependent on the duration, but *not* on the potential, of the depolarizing voltage step that evokes $I_{NaT}$ (*Figure 3H,I*).

One potential benefit of computational modeling is the ability to dissect out possible mechanisms of channel gating by examining the occupancy of the individual channel states as a function of voltage and time. Taking advantage of this benefit, we examined the proportion of Nav channels populating each gating state during a simulated voltage-clamp protocol that evoked $I_{NaR}$ at –45 mV after a 5 ms depolarizing voltage step to 0 mV (*Figure 4*). As illustrated, fast inactivation of $I_{NaT}$ (during the 0 mV step) reflects (simulated) channels exiting the open state and accumulating into the IF1/IF2 states. The activation of $I_{NaR}$ (at –45 mV) reflects channel transitioning from IF1/IF2 back into the open state, and the decay of $I_{NaR}$ during the –45 mV step reflects the time-dependent accumulation of (simulated) Nav channels in the secondary, slow-inactivated state, IS (see *Figures 3A and 4A*). The distinct pathways of inactivation are separated in time, but are not distinguished by differing voltage dependences. Additionally, *Figure 4—figure supplement 1* shows that simulated channels follow a similar inactivation pathway during a sustained depolarization to 0 mV, with an initial accumulation in the IF1/IF2 states and subsequent accumulation in the IS state, a property of the model that reveals why $I_{NaR}$ amplitudes are reduced as the duration of the depolarizing voltage step (that evokes $I_{NaT}$) is increased (*Figure 3F and H*).

## $I_{NaT}$ and $I_{NaR}$ are differentially sensitive to entry into the slow-inactivated state

The simulations (*Figure 4*) indicate the fast decay of $I_{NaT}$ and the much slower decay of $I_{NaR}$ reflect separate, that is, fast and slow, pathways of Nav channel inactivation and, in addition, that the decay of $I_{NaR}$ reflects Nav channels accumulating in an absorbing, slow-inactivated state (i.e., the IS state in *Figure 3*), suggesting there may be discrete Nav channel populations that accumulate in the 'IS' state during prolonged depolarizations. To test this hypothesis directly, we measured peak $I_{NaT}$ and $I_{NaR}$ amplitudes, recorded at 0 and –45 mV, respectively, during sequential voltage-clamp protocols separated by a brief (20 ms) interval at –90 mV; the paradigm is illustrated in *Figure 5A* below the current records. The 20 ms interval at –90 mV between the sequential protocols was determined to be sufficient for the near complete recovery of $I_{NaT}$ from fast inactivation (*Ransdell et al., 2017*; *Aman and Raman, 2007*). However, if the channels that underlie $I_{NaR}$ have accumulated in the second, slow-inactivated (IS) state during the first –45 mV voltage step and recovery from this state is also slow,

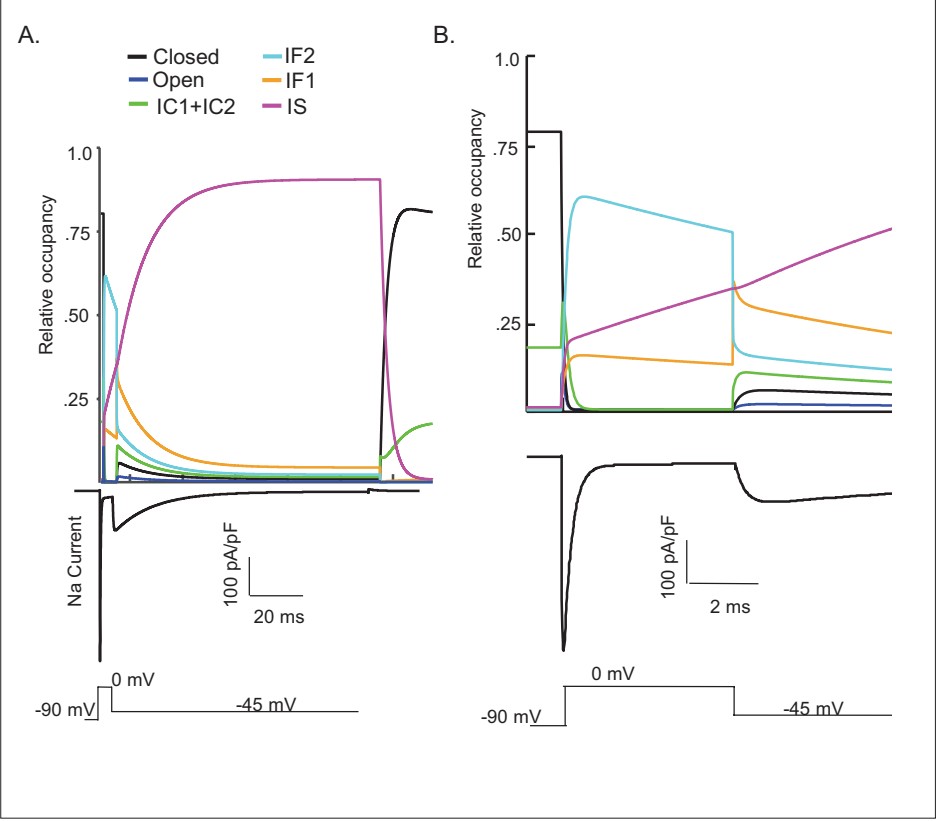

**Figure 4.** Kinetic state transitions during voltage-clamp simulations that evoke voltage-gated sodium current ($I_{NaR}$). There are two parallel pathways of voltage-gated sodium (Nav) channel inactivation (IF1/IF2 and IS) in the novel Markov kinetic state model developed here (*Figure 3*). The occupancies of these states and of the other (i.e., closed, open, etc.) channel gating states during a simulated voltage-clamp protocol, in which $I_{NaR}$ is revealed on membrane hyperpolarization to –45 mV following a brief (5 ms) depolarizing voltage step to 0 mV from a holding potential of –90 mV, are plotted as a function of time in (**A**). The simulated voltage-clamp records and the experimental paradigm are illustrated below the gating state occupancy plot. Expanded (in time) views of the gating state occupancies and the simulated Nav currents are presented in (**B**). In the gating state occupancy plots, *black* represents the closed state, *blue* represents the open state, *green* represents the IC1+ IC2 states, *aqua* represents the IF2 state, *orange* represents the IF1 state, and *purple* represents the IS state.

The online version of this article includes the following figure supplement(s) for figure 4:

**Figure supplement 1.** Simulations reveal that, similar to hyperpolarizing voltage steps, prolonged depolarization results in the accumulation of voltage-gated sodium (Nav) channels in the IS state.

one would expect to see differential effects on peak $I_{NaT}$ and peak $I_{NaR}$ amplitudes when the time interval at –90 mV is sufficient to allow complete recovery of $I_{NaT}$, but too short to allow the complete recovery of $I_{NaR}$. As illustrated in the representative records shown in *Figure 5A*, this voltage-clamp paradigm revealed that, when the time interval at –90 mV was reduced to 20 ms, the amplitude of $I_{NaR}$ was indeed reduced to a greater extent than the amplitude of $I_{NaT}$. Recordings from five additional Purkinje neurons using this voltage-clamp paradigm yielded similar results. Plotting the relative peak amplitudes of $I_{NaT}$ and $I_{NaR}$ measured during the second protocol, compared with the first, reveals that the 20 ms hyperpolarizing voltage step to –90 mV was sufficient to provide nearly complete (0.95 ± 0.01; n = 6) recovery of $I_{NaT}$ (*Figure 5B*), whereas there was a marked reduction in the amplitude of $I_{NaR}$ (0.63 ± 0.05; n = 6), measured during the second, compared with the first, protocol (*Figure 5B*).

## $I_{NaR}$ reflects the transitioning of fast-inactivated Nav channels into an open conducting state

The results presented in *Figure 5* indicate that Nav channels open during membrane hyperpolarizations from depolarized potentials and that these (open) channels recover from inactivation at a rate

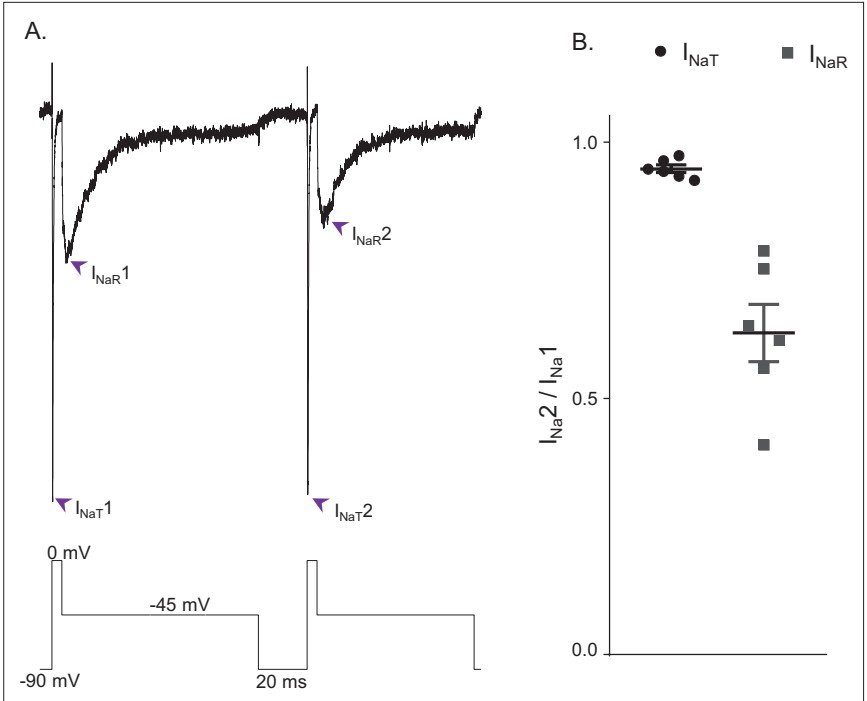

**Figure 5.** $I_{NaR}$ and $I_{NaT}$ display distinct rates of recovery from inactivation. (**A**) Representative voltage-gated sodium (Nav) currents recorded in a mouse cerebellar Purkinje neuron during a voltage-clamp paradigm designed to determine if the relative rates of recovery from inactivation of $I_{NaT}$ and $I_{NaR}$ are distinct. Inward Nav currents were recorded during sequential and identical voltage-clamp steps (to 0 mV for 5 ms and to –45 mV for 100 ms), separated by a brief (20 ms) hyperpolarizing step to –90 mV; the voltage-clamp paradigm is shown below the current records. As is evident, the amplitude of $I_{NaR}$ (at –45 mV) during the second voltage-clamp step to –45 mV was attenuated more than $I_{NaT}$ (during the second step to 0 mV). Similar results were obtained in five additional Purkinje neurons using the voltage-clamp paradigm shown. (**B**) Plot of the relative peak $I_{NaT}$ (circles) and peak $I_{NaR}$ (squares) amplitudes measured during the second voltage-clamp steps (to 0 and –45 mV), compared with the first. As is evident, the relative amplitude of $I_{NaR}$ is reduced (0.63 ± .05; n = 6) to a greater extent (paired Student's t-test; p = .00039) than $I_{NaT}$ (0.95 ± 0.01; n = 6). The mean ± SEM (n = 6) relative $I_{NaT}$ and $I_{NaR}$ amplitudes are also indicated.

that is distinct from the complement of Nav channels responsible for $I_{NaT}$. To determine the relationship between $I_{NaR}$ and the channels that give rise to the persistent component of the sodium current, $I_{NaP}$, we used two voltage-clamp protocols, designed to allow direct measurements of $I_{NaP}$ alone or $I_{NaP}$ plus $I_{NaR}$. In the first protocol, a slow (dV/dt = 0.12 mV/ms) depolarizing voltage ramp (from –100 to 0 mV) was presented and inward currents, reflecting only $I_{NaP}$, were recorded (*Figure 6A*, *blue*). In the same cell, we also recorded Nav currents evoked during a slow (dV/dt = 0.12 mV/ms) hyperpolarizing (from 0 to –100 mV) voltage ramp (*Figure 6A*, *red*). In the latter case, the measured inward currents reflect both $I_{NaP}$ and $I_{NaR}$, that is, channels capable of recovering from a fast-inactivated state into an open (conducting) state on membrane repolarization. It should be noted that $I_{NaR}$ decays (see *Figure 2B*) during the hyper-polarizing voltage-ramp and that the relative amplitudes of $I_{NaR}$ and $I_{NaP}$ to the measured currents vary as function of time during the ramp. In addition, we recorded the currents evoked during depolarizing voltage steps to various test potentials between –75 and +10 mV from a holding potential of –100 mV, and we measured the amplitudes of $I_{NaP}$ directly, at 25 ms after the onset of each depolarizing voltage step (*Figure 6A*, *green*). The current-voltage plots, derived from the data obtained in these experiments, are presented in *Figure 6B*; the colors correspond to those used to illustrate the current records presented in *Figure 6A*. As is evident, the voltage dependences and the magnitudes of the Nav currents recorded using the three voltage-clamp protocols (in the same cell) are indeed indistinguishable. Similar results were obtained in recordings from four additional Purkinje neurons (see Discussion).

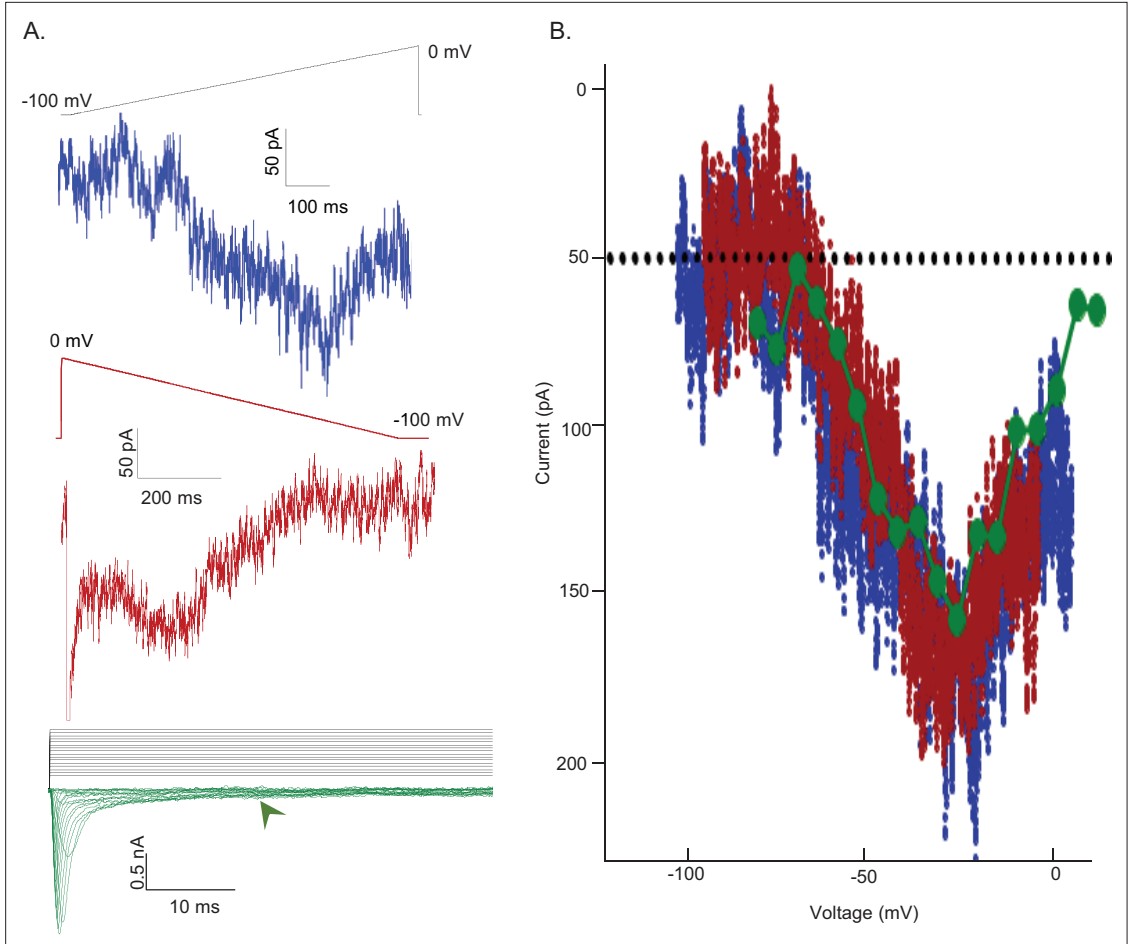

**Figure 6.** Voltage dependences of activation of I$_{NaR}$ and I$_{NaP}$ are indistinguishable. (**A**) To test the hypothesis that non-inactivating voltage-gated sodium (Nav) channels underlie I$_{NaR}$, a depolarizing voltage ramp (*blue*) protocol (from –100 to 0 mV at 0.12 mV/ms) and a steady-state voltage step (*green*) protocol (with depolarizations from a holding potential of –100 mV to test potentials ranging from –75 to 10 mV in 5 mV increments) were used to reveal the magnitude and voltage-dependent properties of the non-inactivating (persistent) component of the Nav current, I$_{NaP}$. In addition, a hyperpolarizing voltage ramp (from 0 to –100 mV at 0.12 mV/ms or dV/dt) was used to reveal both I$_{NaR}$ and I$_{NaP}$. Note that as I$_{NaR}$ decays (see *Figure 2B*) during the hyperpolarizing voltage-ramp, the relative amplitudes of I$_{NaR}$ and I$_{NaP}$ vary during the ramp; the sum of the two current components, not the amplitudes of the individual components, therefore, are measured. The three representative records shown were obtained from the same Purkinje neuron. (**B**) The current-voltage relationships, derived from the records presented in (**A**) are plotted (in the corresponding color). From the records shown in the lowest panel of (**A**), the amplitudes of the steady-state inward currents at 25 ms at each test potential are plotted as points (*green*); the current amplitudes determined (at 2 ms intervals) from the ramp protocols (*red* and *blue* traces) are also plotted. As is evident, the current-voltage relations of the Nav currents recorded using the three different voltage-clamp protocols overlap; the magnitudes of the inward Nav currents are also indistinguishable. Similar results were obtained in four additional Purkinje neurons.

## Nav channel gating model with an OB state does not reproduce the voltage-clamp data

The novel Markov model developed here (*Figure 3A*) is quite different from the previously proposed model of Nav channel gating in mouse cerebellar Purkinje neurons (*Raman and Bean, 2001*). In this earlier model (illustrated in *Figure 7A*), there are two distinct competing pathways that depopulate the open state, one of which involves fast Nav channel inactivation and results in the population of the I6 state (*Figure 7A*) and the other, competing, pathway involves the blockade of open Nav channels and the generation of the OB state (*Figure 7A*). The isolation of the OB state from all of the other kinetic states except the open state is a distinctive feature in the model of Raman and Bean (*Figure 7A*). This configuration means that channels that have entered the OB state can only exit this state (i.e., become unblocked) by transitioning into the open/conducting state to generate resurgent Na+ influx, that is, I$_{NaR}$ (*Raman and Bean, 2001*). Although it was suggested that the blocking particle

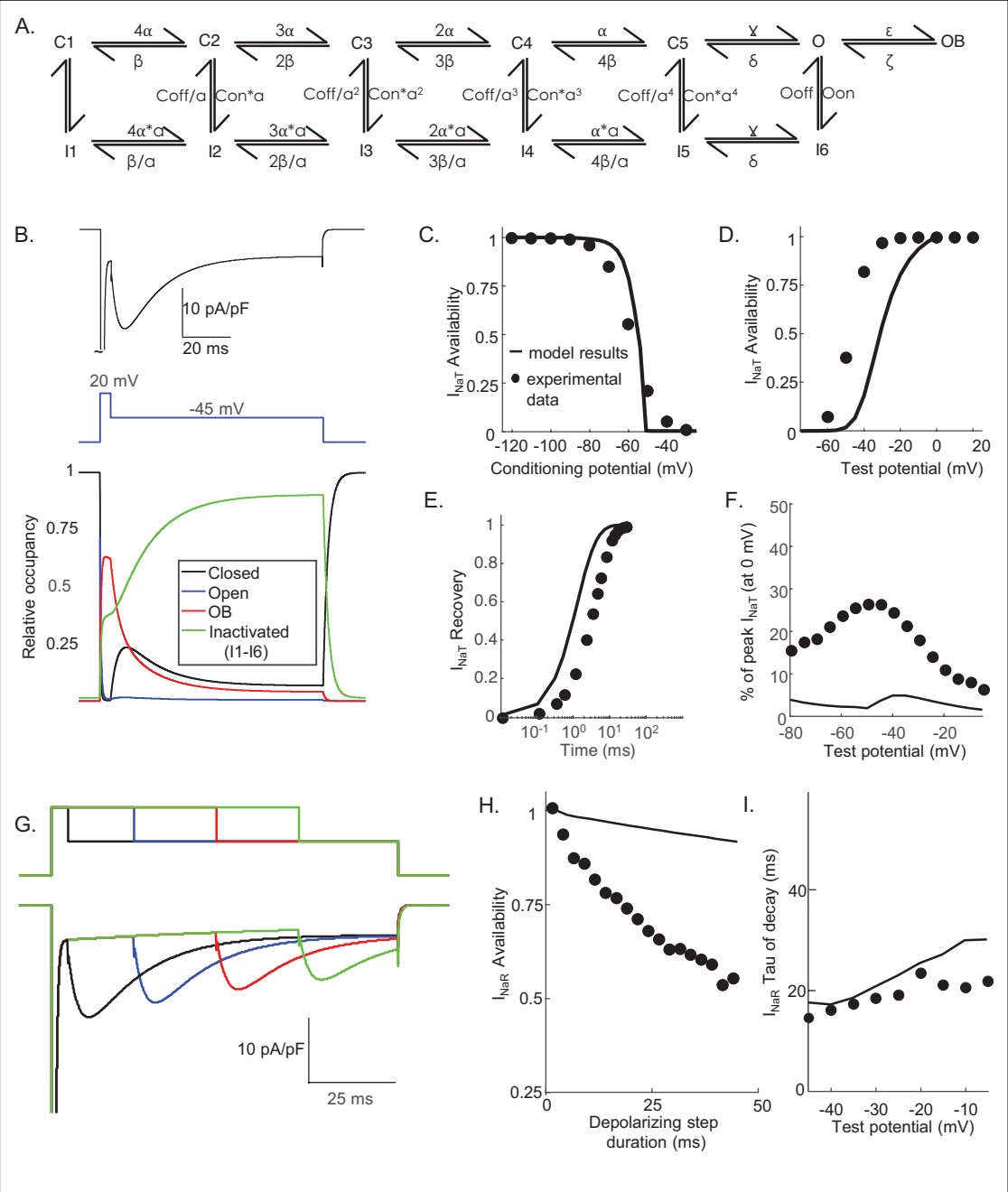

**Figure 7.** Simulations using the Raman-Bean model of voltage-gated sodium (Nav) channel gating do not recapitulate the acquired voltage-clamp data. (**A**) The previously described Markov kinetic state model of Nav channel gating in mouse cerebellar Purkinje neurons (**Raman and Bean, 2001**) is illustrated. (**B**) Representative simulated inward Nav current waveforms, produced by this (**A**) model using the voltage-clamp paradigm shown below the current records, are presented. The time-matched normalized occupancies of the combined closed (C1–C5, shown in *black*), open (O, shown in *blue*), open-blocked (OB, shown in *red*), and combined inactivated (I1–I6, shown in *green*) gating states are plotted below the voltage protocol. (**C–E**) Comparisons of the time- and voltage-dependent properties of $I_{NaT}$ derived from simulations using the model in (**A**) with (our) experimental data obtained in recordings from mouse cerebellar Purkinje neurons (the same data as were used to generate the results in **Figure 3**); filled circles represent the mean experimental data and the lines represent the results of the simulations. (**F**) Relative $I_{NaR}$ amplitudes (normalized to peak $I_{NaT}$ at 0 mV) are plotted as a function of the hyperpolarizing test potential. (**G**) Simulated $I_{NaR}$ waveforms, produced on membrane hyperpolarizations to –45 mV following depolarizing voltage steps to +20 mV of varying durations, are shown. (**H**) Peak normalized $I_{NaR}$ amplitudes (at –45 mV), derived from the simulations in (**G**), are plotted as a function of the duration of the prior +20 mV depolarizing voltage step (solid line), together with the mean experimental data (filled circles) obtained in recordings from mouse cerebellar Purkinje neurons (the same data as used to generate the results in **Figure 3**). (**I**) The kinetics of $I_{NaR}$ decay, derived from single exponential fits to the decay phases of the currents recorded at various membrane potentials, are presented in (**I**); the solid

*Figure 7 continued on next page*

*Figure 7 continued*

lines indicate the results of the simulations, and the filled circles are the mean experimental data obtained in recordings from mouse cerebellar Purkinje neurons (the same data as used to generate the results in *Figure 3*).

responsible for producing the OB state was a protein, specifically the Nav channel accessory subunit Navβ4 (*Grieco et al., 2002*; *Grieco et al., 2005*; *Bant and Raman, 2010*), it was subsequently demonstrated that $I_{NaR}$ is reduced, but is *not* eliminated, in cerebellar Purkinje neurons in (*Scn4b$^{-/-}$*) mice lacking Navβ4 (*Ransdell et al., 2017*; *White et al., 2019*) (see Discussion).

Subsequent efforts here were focused on determining directly whether the Nav channel gating model in which $I_{NaR}$ is generated by the OB mechanism (*Figure 7A*) could/would also reliably reproduce the detailed time- and voltage-dependent properties of the Nav currents determined experimentally in mouse cerebellar Purkinje neurons (and presented in *Figures 1, 2, 5, and 6*). As illustrated in *Figure 7B*, simulations with the OB model for $I_{NaR}$ generation (*Figure 7A*) provided transient and resurgent Nav current components (*Figure 7B*, upper panel) that resemble those measured experimentally (*Figure 1A*). A time-locked plot of gating state occupancies with this model reveals that $I_{NaR}$ activation occurs as simulated channels exit the OB state and into the open (conducting) state (*Figure 7B*, lower panel). Additional simulations revealed that this model also recapitulates the voltage and time dependences of $I_{NaT}$ activation, inactivation and recovery from inactivation (*Figure 7C–E*) in mouse cerebellar Purkinje neurons, although the modeled $I_{NaT}$ activates at more hyperpolarized voltages than native $I_{NaT}$. Although recapitulating the voltage dependence of $I_{NaR}$ activation (*Figure 7F*), the model also predicts that the magnitude of $I_{NaR}$ evoked at all hyperpolarized membrane potentials, relative to the peak amplitude of $I_{NaT}$ (evoked at 0 mV), is much smaller than observed experimentally (*Figure 7F*). It should be noted, however, that the relative $I_{NaR}$ amplitudes measured here were obtained in experiments conducted with a much higher (151 mM) extracellular Na$^+$ concentration than the concentration (of 50 mM) used in earlier studies (*Raman and Bean, 2001*), which likely accounts for some (if not all) of the difference in the relative magnitudes of $I_{NaR}$ measured here and previously (*Raman and Bean, 2001*).

In the open-channel block model of $I_{NaR}$ gating (*Figure 7A*), more positive depolarizing voltage steps promote entry into the OB state, whereas channels are favored to undergo fast (conventional) inactivation at more hyperpolarized membrane potentials (*Raman and Bean, 2001*; *Lewis and Raman, 2014*). When the voltage-clamp protocols used to generate the data presented in *Figure 2* were used in simulations with the open-channel block model (*Figure 7A*), however, substantial differences between the predictions of this model and our experimental data were revealed. In contrast to what is observed experimentally (*Figure 2*), for example, increasing the duration of the brief (5 ms) depolarizing voltage step (to 0 mV) in the open channel block model resulted in very little time-dependent attenuation of the amplitude of $I_{NaR}$ recorded on membrane hyperpolarization to –45 mV (*Figure 7G and H*). In addition, the time course of the decay of the resurgent currents predicted by the open channel block model (*Figure 7A*) are slower than we observed experimentally for $I_{NaR}$ in mouse cerebellar Purkinje neurons (*Figure 7I*).

## The fast- and slow-inactivated Nav channel states are populated separately and at different rates

The experimental data and the simulations using the novel Nav channel gating model developed here (*Figure 3A*) suggest that $I_{NaR}$ is mediated by Nav channels transiting from a fast-inactivated state (IF1) into the open state, and subsequently accumulating in an absorbing, slow-inactivated state (IS). It is also possible, however, that the absorbing slow-inactivated state that underlies $I_{NaR}$ decay reflects Nav channels accumulating into a long-term inactivated state, that is, a state in which channels are non-conducting for hundreds of ms, for example, by a blocking particle(s) that competes on a time scale similar to conventional fast inactivation (*Dover et al., 2010*; *Venkatesan et al., 2014*). To test this possibility directly, a voltage-clamp protocol was developed to allow direct comparison in the same cell of $I_{NaR}$ recorded during a single (80 ms) hyperpolarizing voltage step to –45 mV, presented following a brief (5 ms) depolarization to 0 mV, with $I_{NaR}$ recorded at –45 mV during successive brief (2 ms) hyperpolarizing voltage steps interspersed with brief (5 ms) depolarizations to 0 mV (*Figure 8A*). If a competing extrinsic blocking particle has fast-onset, competes with conventional inactivation and is absorbing, one would expect to see reductions in the amplitudes of $I_{NaR}$ recorded during

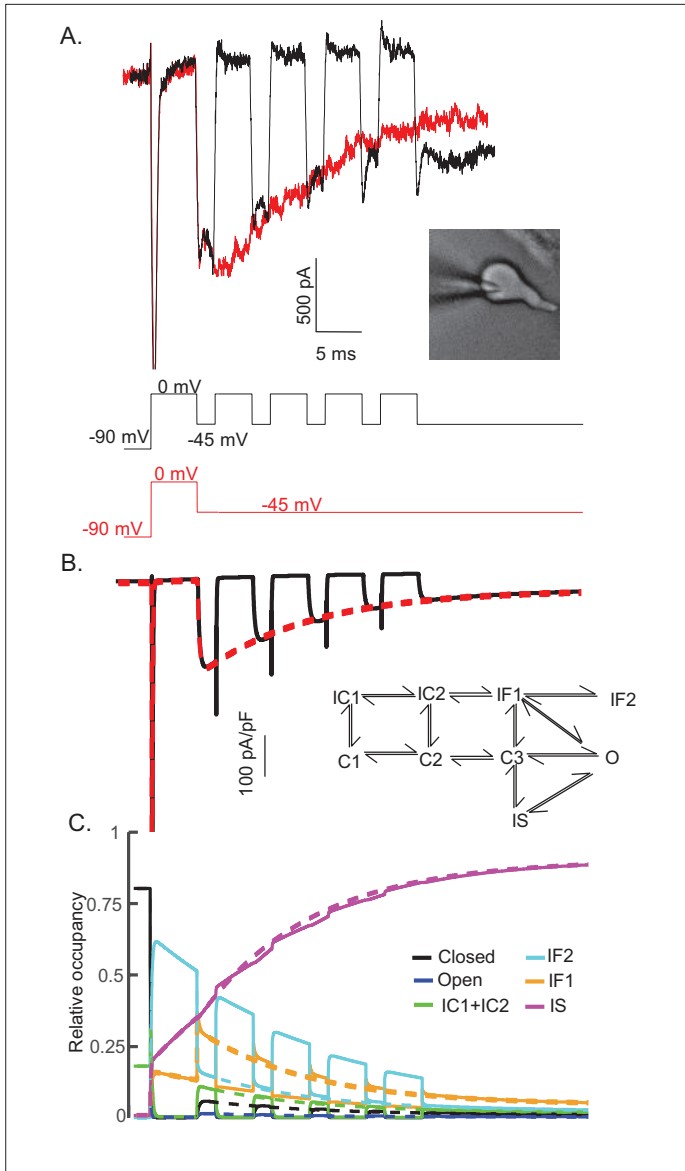

**Figure 8.** The time course and amplitude of $I_{NaR}$ are recapitulated during repetitive brief depolarizing steps. To determine if two competing inactivation states underlie the observed differences in $I_{NaT}$ and $I_{NaR}$ recovery from inactivation (illustrated in **Figure 5**), a protocol was developed to allow direct comparison of $I_{NaR}$ recorded during a single (80 ms) hyperpolarizing voltage step to –45 mV (*red*), presented following a brief (5 ms) depolarization to 0 mV, with $I_{NaR}$ recorded (in the same cell) at –45 mV during successive brief (2 ms) hyperpolarizing voltage steps interspersed with brief (5 ms) depolarizations to 0 mV (*black*). Representative records are shown in (**A**); the voltage-clamp paradigms are illustrated below the current records. Similar results were obtained in four additional Purkinje neurons. As is evident (**A**), the envelope of the currents generated using these two protocols superimpose, suggesting that the inactivation pathway responsible for $I_{NaR}$ decay does not compete with fast inactivation. (**B**) Simulated current waveforms, generated using the same two voltage-clamp protocols illustrated in (**A**) with the novel kinetic state model presented in **Figure 3A**, are shown. (**C**) Gating state occupancies for simulated current traces are shown with *black* representing the closed state, *blue* representing the open state, *green* representing the IC1+ IC2 states, *aqua* representing the IF2 state, *orange* representing the IF1 state, and *purple* representing the IS state. For direct comparison of the results of the simulations using the voltage-clamp protocols illustrated in (**A**) with the Raman-Bean gating model (2001), see **Figure 8—figure supplement 1**.

The online version of this article includes the following figure supplement(s) for figure 8:

**Figure supplement 1.** Gating state occupancies/transitions produced in the Raman-Bean model (with the voltage-clamp protocols used in **Figure 8**).

each successive hyperpolarizing voltage step to –45 mV compared with $I_{NaR}$ recorded during a sustained hyperpolarizing voltage step to –45 mV. As is evident in the experimental records shown in *Figure 8A*, however, $I_{NaR}$ waveforms evoked (in the same cell) using these two voltage-clamp protocols were quite similar.

The representative traces presented in *Figure 8A* were recapitulated in simulations (*Figure 8B*) using the novel gating state model (*Figure 3A*) developed here. The current waveforms generated by the model are indistinguishable from the experimental results (compare *Figure 8A and B*). In addition, and without tuning any of the model parameters, the kinetic state occupancy plots generated using the two voltage-clamp protocols were also very similar (*Figure 8C*). This voltage-clamp protocol was also applied in simulations using the previously described (*Figure 7A*) open channel block model (*Raman and Bean, 2001*). In this case, in marked contrast with the results presented in *Figure 8*, the model does not reproduce the experimental data (*Figure 8—figure supplement 1A*). The simulations revealed that, in this model, channels did not maximally enter into the OB state on depolarizations to 0 mV and the 2 ms hyperpolarizations were not sufficient to fully activate $I_{NaR}$ (*Figure 8—figure supplement 1B*). Additionally, in this model, during each of the successive depolarizations (to 0 mV), transient Nav currents were revealed (*Figure 8—figure supplement 1A*), reflecting channels exiting the OB state on membrane hyperpolarization and re-entering the OB state on membrane depolarization.

## Simulating $I_{NaR}$ in Scn4b-/- cerebellar Purkinje neurons

We previously reported that the targeted deletion of *Scn4b* in mice results in a marked (~50%) reduction in $I_{NaR}$ amplitudes in cerebellar Purkinje neurons (*Figure 9A*) without measurable effects on $I_{NaR}$ kinetics or voltage dependence (*Ransdell et al., 2017*). In subsequent studies conducted using a different *Scn4b$^{-/-}$* mouse line, it was reported that the targeted deletion of *Scn4b* had no effects on $I_{NaR}$ kinetics, voltage dependences, or amplitudes (relative to $I_{NaT}$) (*White et al., 2019*; *Ransdell et al., 2017*). To explore the ability of the novel model of Nav gating, developed and presented here (*Figure 3A*), to scale the amplitude of $I_{NaR}$ while leaving the time- and voltage-dependent properties of the currents unaffected,

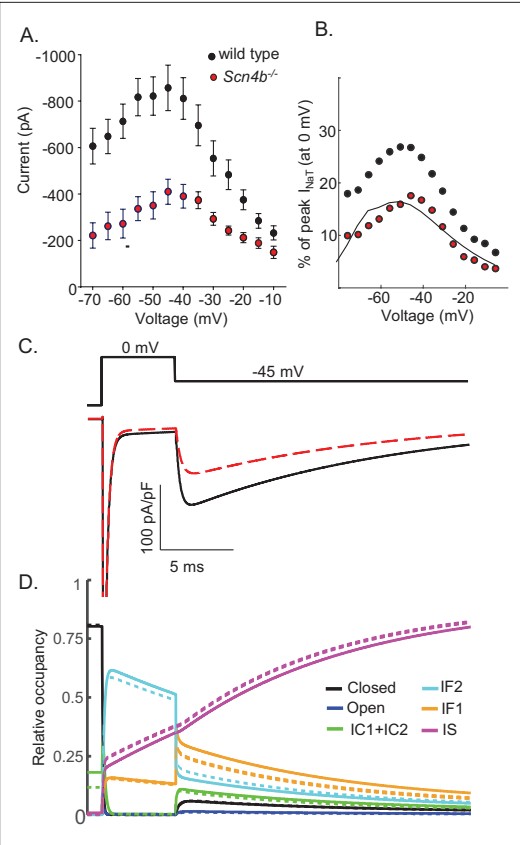

**Figure 9.** Promoting entry into the slow-inactivated state reduces voltage-gated sodium current ($I_{NaR}$) amplitudes. (**A**) Mean ± SEM peak $I_{NaR}$ amplitudes, measured on membrane hyperpolarizations following brief depolarizing voltage steps to +10 mV, in wild type (*black*) and *Scn4b$^{-/-}$* (*red*) mouse cerebellar Purkinje neurons are plotted as a function of membrane voltage are shown (data were reproduced with permission from *Ransdell et al., 2017*). Peak $I_{NaR}$ amplitudes in individual wild type and *Scn4b$^{-/-}$* cells were also normalized to peak $I_{NaT}$ measured (at 0 mV) in the same cell, and the mean $I_{NaR}$ as a percentage of peak $I_{NaT}$ in wild type (*black*) and *Scn4b$^{-/-}$* (*red*) cells are plotted (as points) in (**B**); the solid line is the normalized relative $I_{NaR}/I_{NaT}$ generated by the *Scn4b$^{-/-}$* model. (**C**) Consistent with the experimental data, the kinetics of $I_{NaR}$ are not affected measurably by the loss of *Scn4b* (Navβ4) in the model, whereas $I_{NaR}$ amplitudes are reduced to ~50% of wild type $I_{NaR}$ levels (**C**). A time-locked plot of the gating state transitions (**D**) indicates that $I_{NaR}$ amplitudes are reduced in the *Scn4b$^{-/-}$* model (dashed lines) due to a decrease in IF2 occupancy and an increase in IS occupancy. In this gating state occupancy plot, *black* represents the closed state, *blue* represents the open state, *green* represents the IC1+ IC2 states, *aqua* represents the IF2 state, *orange* represents the IF1 state, and *purple* represents the IS state.

we optimized the parameters of the model (see Materials and methods) to reproduce the experimentally determined reduction in $I_{NaR}$ amplitudes with the loss of Navβ4 (*Ransdell et al., 2017*; *White*

*et al., 2019*). As illustrated in *Figure 9B*, although reduced in amplitude, the voltage dependence of $I_{NaR}$ generated by the *Scn4b$^{-/-}$* Nav channel gating model is very similar to wild type $I_{NaR}$. In *Figure 9C*, representative $I_{NaR}$ waveforms generated by the gating models of wild type and *Scn4b$^{-/-}$* Nav currents are superimposed; the time courses of wild type and *Scn4b$^{-/-}$* $I_{NaR}$ are indistinguishable. Time-locked with the wild type and *Scn4b$^{-/-}$* Nav current traces are plots of the Nav channel gating state occupancies as a function of time (*Figure 9D*) in the wild type (solid lines) and the *Scn4b$^{-/-}$* (dashed lines) $I_{NaR}$ models. The gating state occupancy plots (*Figure 9D*) reveal that the attenuation of the amplitude of $I_{NaR}$ in the *Scn4b$^{-/-}$* model is the result of the reduced accumulation of channels in the IF2 state during the initial depolarization and increased occupancy in the IS state. Together, these data suggest that Navβ4 delays entry into the slow-inactivated state (IS), allowing for greater recovery from conventional inactivation and thus, larger $I_{NaR}$ amplitudes (see Discussion).

## Discussion

Using a combined experimental and modeling approach, we describe here a novel mechanism for $I_{NaR}$ gating that reflects two kinetically distinct (fast and slow) pathways of Nav channel inactivation. Importantly, the model reliably recapitulates the detailed time- and voltage-dependent properties of $I_{NaR}$ determined experimentally in mouse cerebellar Purkinje neurons. The model, for example, accounts for the experimental finding that the peak amplitude of $I_{NaR}$ (recorded on membrane hyperpolarizations following brief depolarizations that evoke $I_{NaT}$) is sensitive to the duration of the preceding depolarizing voltage step, with longer depolarizations resulting in lower $I_{NaR}$ amplitudes. Importantly, the dependence of $I_{NaR}$ amplitudes on the duration of the depolarizing voltage step was also observed when the inward driving force of Na$^+$ was reduced or eliminated. In addition, the model reproduces the experimental finding that $I_{NaR}$ amplitudes are not dependent on the voltage of the initial depolarizing voltage step (that evokes $I_{NaT}$). Finally, and also consistent with the experimental data, the voltage dependence of $I_{NaR}$ (revealed on membrane hyperpolarizations) mirrors the voltage dependence of $I_{NaP}$ (see *Figure 6*). Interestingly, this last observation, that is, that the voltage dependence of $I_{NaP}$ is identical to the voltage dependence of $I_{NaR}$, was previously reported (*Kay et al., 1998*).

The model developed here (*Figure 3A*) is distinct from a previously proposed model (*Figure 7A*) of Nav channel gating in mouse cerebellar Purkinje neurons (*Raman and Bean, 2001*) that involves two competing pathways out of the open state, that is, channels transitioning into either an OB state or an inactivated state. In this earlier model (*Figure 7A*) of $I_{NaR}$ gating, a blocking particle occludes the Nav channel pore that is opened on membrane depolarization, functionally competing with conventional, fast inactivation, and, in addition, the OB state is isolated off the open/conducting state (*Raman and Bean, 2001*). In this model, entry into the OB state is promoted by membrane depolarizations to positive potentials and, on subsequent membrane hyperpolarization, the blocking particle is expelled by the Na$^+$ driven through the unblocked Nav channel pore (*Aman and Raman, 2010*). In addition, the larger the driving force on Na$^+$, the more rapidly the blocker is displaced (*Aman and Raman, 2010*). This process, while intrinsically voltage-independent, therefore, is tied to the driving force on Na$^+$, consistent with the experimental observation that the peak amplitude of $I_{NaR}$ is observed at relatively hyperpolarized (−45 to –30 mV) membrane potentials (*Aman and Raman, 2010*; *Lewis and Raman, 2014*). At more negative membrane potentials, closed state inactivation begins to dominate, and $I_{NaR}$ is reduced (*Raman and Bean, 2001*; *Lewis and Raman, 2014*). The experimental results presented here, however, reveal that the peak amplitude of $I_{NaR}$ is not affected by the voltage of the initial membrane depolarization (*Figure 2D*), but rather *is affected by the duration* of the prior depolarizing voltage step (*Figure 2A and B*). The finding that the magnitude of $I_{NaR}$ is substantially reduced when the initial depolarizing voltage step is increased in duration was previously reported (*Raman and Bean, 2001*). In addition, experimental results presented here demonstrate that the time-dependent accumulation of Nav channels into the slow-inactivated state occurs independent of the direction of the movement of permeating Na$^+$ ions.

### Recovery from conventional Nav channel inactivation into an open/ conducting state

The experiments and the simulations presented here suggest that $I_{NaR}$ reflects the transitioning of Nav channels that have undergone conventional fast inactivation, into an open/conducting state on

membrane hyperpolarizations. Consistent with the slow decay of $I_{NaR}$, the model includes two parallel, but distinct, inactivation pathways: the fast (i.e., IF1 and IF2) inactivation pathway and the slow (i.e., IS) inactivation pathway, which satisfies the key experimental finding that the amplitude of $I_{NaR}$ is insensitive to the voltage of the initial depolarization (*Figure 2*). Although it was previously reported that the slower component of the biexponential decay of $I_{NaT}$ (measured at –30 mV) in Purkinje neurons is identical to the decay rate of $I_{NaR}$ (also measured at –30 mV), this observation was interpreted as suggesting that all Nav channels initially undergo open channel block (*Raman and Bean, 2001*). The results of the experiments here involving prolonged depolarizations (*Figure 2A–C*), however, are not consistent with this model. In the experiments here, we found that there is still a time-dependent attenuation of $I_{NaR}$ amplitudes on membrane hyperpolarizations following depolarizations to positive membrane potentials, with little or no inward driving force on $Na^+$. These findings are inconsistent with the open-channel block hypothesis, in which positive membrane potentials are thought to promote and stabilize open-channel block (*Raman and Bean, 2001*; *Aman and Raman, 2010*; *Lewis and Raman, 2014*).

One assumption usually made when considering the gating of Nav channels is that deactivation must occur prior to recovery from inactivation. Or, to state another way, Nav channels cannot exit the conventional inactivated state directly into an open or conducting state. This idea is based on voltage-clamp experiments conducted on CA1 hippocampal neurons (*Kuo and Bean, 1994*) and the squid giant axon (*Armstrong, 2006*), which revealed that recovery from inactivation occurs with a delay and is voltage-dependent. Indeed, if recovery from inactivation requires all four of the VSDs of domains I–IV to move into the deactivated position, the model presented here is not possible because channels that have undergone conventional, fast inactivation would not be able to move directly into an open/conducting state without first transiting through an intermediate closed state. However, if deactivation of VSD IV is sufficient to release the cytosolic DIII–DIV linker peptide from the channel pore, while VSD I, II, and III remain in the activated conformation, Nav channels could transit from an inactivated state directly into an open/conducting state. There are several reports suggesting that this is possible. *Schiavon et al., 2012*, for example, demonstrated that scorpion beta toxins induce $I_{NaR}$ in heterologously (in HEK-293 cells) expressed Nav channels. These toxins were previously shown to cause a negative shift in the voltage dependence of DI-VSD, DII-VSD, and DIII-VSD activation by trapping the DII-VSD in the activated position after depolarizing prepulses (*Cestèle et al., 1998*; *Schiavon et al., 2006*). Thus, on repolarization, the channel recovers from inactivation, allowing resurgent $Na^+$ influx.

Resurgent Nav currents have also been induced in heterologously expressed (in HEK-293 cells) Nav1.5-encoded channels following application of the classical type II pyrethroid, deltamethrin. Similar to our model for the generation of $I_{NaR}$, the deltamethrin-modified Nav1.5 channels were found to recover from inactivation prior to deactivation (*Thull et al., 2020*). In addition, it has also been reported (*Cummins et al., 2004*) that heterologously expressed (in HEK-293 cells) mutant Nav1.7-encoded channels have prolonged deactivation time constants at the hyperpolarized membrane potentials associated with the activation of $I_{NaR}$. These mutant Nav1.7 channels were also reported to display an increase in the relative amplitude of the non-inactivating Nav current (i.e., $I_{NaP}$). These two observations are clearly consistent with the $I_{NaR}$ gating model proposed here in which Nav channels can recover from conventional, fast inactivation directly into an open/conducting state. It is, however, worth noting that in this report (*Cummins et al., 2004*), $I_{NaR}$ was never measured directly, instead the membrane was repolarized prior to the completion of $I_{NaT}$ inactivation. As a result, it is not possible to conclude, with certainty, that the mutant Nav1.7 channels were recovering from inactivation into an open/conducting state, or simply displaying prolonged deactivation at hyperpolarized membrane potentials.

In developing the model (*Figure 3A*), we found that two fast-inactivated states, that is, IF1 and IF2, were required to recapitulate the experimental data. It is certainly possible, however, that IF1 and IF2 are substrates of the same channel state. We also note here that, initially, the model was developed with two slow-inactivated states (IS, IS2) for symmetry. The simulations revealed, however, that the two slow-inactivated states were redundant and that we were able to fit the experimental data reliably with only one of these slow-inactivated states.

It is important to note that we appreciate that our model topology and rate constants, while reliably recapitulating the time- and voltage-dependent properties of the currents determined experimentally, are not necessarily 'unique' or potentially the 'most simple'. Indeed, in using Markov kinetic

state modeling, it is not possible to tell the uniqueness or the ambiguity of the model (both with regard to the parameters and the model topology). Following the approach of *Menon et al., 2009*, using a state mutating genetic algorithm to vary topologies in a Markov model, *Mangold et al., 2021*, recently reported the development of an algorithm to enumerate all possible model structures distinctly using rooted graph theory (e.g., all possible combinations of models, rooted around a single open state). These efforts revealed that there are many model structures and parameter sets that can adequately fit some experimental datasets.

## Molecular determinants of I$_{NaR}$

Soon after the discovery of I$_{NaR}$, it was reported that the targeted deletion of *Scn8a* (which encodes the Nav1.6 α subunit) in mouse cerebellar Purkinje neurons resulted in a 90% reduction in the amplitude of I$_{NaR}$, suggesting that, of the α subunits (Nav1.1 and Nav1.6) expressed in Purkinje neurons (*Vega-Saenz de Miera et al., 1997*; *Xiao et al., 2013*), Nav channels formed by Nav1.6 are the major contributors to I$_{NaR}$ (*Raman et al., 1997*). However, it was later reported that the targeted deletion of *Scn1a* (which encodes the Nav1.1 α subunit) also results in ~65% reduction in I$_{NaR}$, suggesting that multiple Nav α subunits contribute to I$_{NaR}$ in cerebellar Purkinje neurons and, in addition, that the contributions are non-linear. To date, all but one (Nav1.3) of the nine Nav channel α subunits have been shown to mediate, or can be induced to mediate, I$_{NaR}$ in native or heterologous cells (*Lewis and Raman, 2014*; *Jarecki et al., 2010*, *Do and Bean, 2004*; *Tan et al., 2014*). It is unclear whether Nav1.3-encoded Nav channels also generate I$_{NaR}$, as it appears that voltage-clamp studies focused on exploring this possibility have not been conducted to date.

In the model proposed here, the amplitude of I$_{NaR}$ is dependent on, and ultimately regulated by, two factors: the proportion of Nav channels that are non-inactivating at hyperpolarized voltages, that is, the number of channels that recover from inactivation into an open state on repolarization; and, the proportion of these non-inactivating channels that accumulate into a slow-inactivated state. In the simulations presented in *Figure 9*, we show that by promoting the occupancy of the IS (slow-inactivated) state, we can recapitulate the experimental observation that I$_{NaR}$ amplitudes are reduced in *Scn4b$^{-/-}$*, compared with wild type, cerebellar Purkinje neurons, suggesting the possibility that Navβ4 expression regulates I$_{NaR}$ by influencing slow inactivation. The hypothesis that I$_{NaR}$ is directly affected by Nav channel slow inactivation is also consistent with previous work (*Hampl et al., 2016*), showing that mutations in the Nav1.6 and Nav1.7 α subunits that enhance or inhibit slow inactivation also result in reduced or increased, respectively, I$_{NaR}$ amplitudes. Interestingly, it has also been reported that mutation of the IFM motif (to QQQ) in Nav1.4, in addition to completely eliminating fast inactivation, increases the proportion of channels that enter the slow inactivated state (*Featherstone et al., 1996*), suggesting that the model developed here (*Figure 3A*) in which the fast and slow components of inactivation are exclusive, may be applicable to diverse Nav channels in different cell types and encoded by different α subunits.

There are a number of additional factors, intrinsic and extrinsic, to Nav α subunits, as well as additional pre- and post-translational mechanisms, that regulate persistent Nav currents (*Aman et al., 2009*; *Hammarstrom and Gage, 1998*; *Paul et al., 2016*; *Lin and Baines, 2015*) and slow inactivation (*Chen et al., 2008*; *Silva, 2014*; *Chen et al., 2006*; *Carr et al., 2003*). In the model developed and presented here, the combined effects of these factors/mechanisms will regulate/modulate the amplitudes and the time- and voltage-dependent properties of I$_{NaR}$. In this context, it is interesting to note that the experiments presented in *Figure 5* indicate that the Nav channels underlying I$_{NaR}$, compared to the sum of the Nav channels underlying I$_{NaT}$, are differentially sensitive to slow inactivation. This observation clearly suggests that there is functional (and perhaps molecular) heterogeneity in the population of Nav channels underlying the Nav currents in mouse cerebellar Purkinje neurons, with channels underlying I$_{NaR}$ having properties distinct from at least a portion of the Nav channels that participate in the generation of I$_{NaT}$, but that do not produce I$_{NaR}$. This possibility is consistent with results presented by *White et al., 2019*, in which I$_{NaR}$ was found to have a greater sensitivity to tetrodotoxin block than I$_{NaT}$ in mouse cerebellar Purkinje neurons. It should also be noted that the Markov model developed here was created to reflect a single channel population, that is, Nav channels capable of generating both I$_{NaT}$ and I$_{NaR}$. As such, the developed Markov model does not reproduce the differential recovery from inactivation presented in *Figure 5*.

## Functional implications

Based on the experimental data and the computational modeling presented here, we propose a novel, blocking particle-independent, gating mechanism for the generation of $I_{NaR}$ that involves two, kinetically distinct inactivation pathways. The modeling results suggest that two parameters are critical in determining the magnitude and the time- and voltage-dependent properties of $I_{NaR}$: (1) the relative amplitude of the persistent Nav current, $I_{NaP}$, component; and, (2) the proportion of the persistent Nav channels (channels that fail to undergo fast inactivation) that undergo slow inactivation. Interestingly, $I_{NaR}$ has now been identified in over 20 types of neurons, many of which do not display the high rates of repetitive firing that are characteristics of cerebellar Purkinje neurons, suggesting that the role(s) of $I_{NaR}$ in the regulation of membrane excitability are diverse, and likely distinct, in different neuronal cell types (*Lewis and Raman, 2014*). Recent studies conducted on serotonergic raphe neurons, for example, suggest that the accumulation of Nav channels in a slow-inactivated state functions as a homeostatic brake on repetitive firing (*Navarro et al., 2020*). In addition, $I_{NaR}$ has been implicated in several inherited and acquired neurological diseases (*Lewis and Raman, 2014*), including paroxysmal extreme pain disorder, paramyotonia congenita (*Jarecki et al., 2010*), and chemotherapy-induced neuropathy (*Lewis and Raman, 2014*; *Sittl et al., 2012*), as well as epilepsy (*Hargus et al., 2013*). Given the implications of these findings, it will be of considerable interest to detail the properties of $I_{NaR}$ in different types of neurons and to explore directly the hypothesis that the rate of decay of $I_{NaR}$ on membrane repolarization plays a role in determining how much persistent sodium current is available to contribute to repetitive firing, as well as to define the molecular mechanisms that control $I_{NaR}$ amplitudes, kinetics, and functioning in diverse neuronal cell types.

# Materials and methods

## Animals

All animal experiments were performed in accordance with the guidelines published in the National Institutes of Health Guide for the Care and Use of Laboratory Animals. Protocols were approved by the Washington University Institutional Animal Care and Use Committee (IACUC). Postnatal day 12–16 (P12–P16) male and female C57BL6/J (Jackson laboratories) mice were used in the experiments reported here.

## Isolation of neonatal cerebellar Purkinje neurons

For the preparation of isolated cerebellar Purkinje neurons, postnatal day 12–16 (P12–P16) animals were anesthetized with 1.25% Avertin and brains were rapidly removed and placed in ice-cold isolation medium containing (in mM): 82 $Na_2SO_4$, 30 $K_2SO_4$, 5 $MgCl_2$, 10 HEPES, 10 glucose, and 0.001% phenol red (at pH 7.4). Using a scalpel, the cerebellum was removed, minced into small chunks and incubated in isolation medium containing 3 mg/ml protease XXIV at 33°C for 10–15 min. Following this incubation period, the tissue pieces were washed with enzyme-free isolation medium containing 1 mg/ml bovine serum albumin and 1 mg/ml trypsin inhibitor. The tissue pieces were transferred to artificial cerebral spinal fluid (ACSF) that was continuously bubbled with 95% oxygen/5% carbon dioxide and contained (in mM): 125 NaCl, 2.5 KCl, 1.25 $NaH_2PO_4$, 25 $NaHCO_3$, 2 $CaCl_2$, 1 $MgCl_2$, and 25 dextrose (310 mosmol/l) at 22–23°C. Tissue pieces were triturated with a fire-polished glass pipette. An aliquot of the cell suspension was placed on a coverslip in the recording chamber and superfused with fresh ACSF (at a rate of 0.5 ml/min), saturated with 95% $O_2$/5% $CO_2$, for 25 min before beginning electrophysiological experiments.

## Electrophysiological recordings

Whole-cell voltage-clamp recordings were obtained at room temperature from visually identified cerebellar Purkinje neurons using differential interference contrast optics. Data were collected using a Multiclamp 700B patch clamp amplifier interfaced to a Dell PC with a Digidata 1332 and pCLAMP 10 software (Axon Instruments, Union City, CA). In all recordings, tip potentials were zeroed prior to forming a giga-ohm membrane–pipette seal. Pipette capacitances were compensated using the pCLAMP software. Signals were acquired at 50–100 kHz and filtered at 10 kHz prior to digitization and storage.

In the experiments here, $I_{NaR}$ was routinely recorded in ACSF (containing ~151 mM Na$^+$; see above) with 5 mM tetraethylammonium chloride (TEA-Cl) and 250 µM cadmium chloride (CdCl$_2$) added. To decrease Nav currents, improve the spatial control of the membrane voltage (space-clamp) and enable reliable recordings of $I_{NaT}$ or $I_{NaP}$, $I_{NaR}$ was also recorded in some experiments (see *Figure 6*) in bath solution containing (in mM): 25 NaCl, 100 TEACl, 2.5 KCl, 1.25 NaH$_2$PO$_4$, 25 NaHCO$_3$, 2 CaCl$_2$, 1 MgCl$_2$, 25 dextrose, and 0.25 CdCl$_2$. Recording pipettes routinely contained (in mM): 110 CsCl, 15 TEACl, 5 4AP, 1 CaCl$_2$, 2 MgCl$_2$, 10 EGTA, 4 Na$_2$-ATP, and 10 HEPES, pH 7.25 300 mosmol/l. Alternative bath and internal solutions were used in experiments presented in *Figure 2*; these solutions are described in the Results section and in the legend to *Figure 2*. In all experiments, recording pipettes had resistances of 2–3 MΩ.

Membrane capacitances were determined by analyzing the decays of capacitive currents elicited by short (25 ms) voltage steps (±10 mV) from the holding potential (−70 mV). Whole-cell membrane capacitances ($C_m$) were calculated for each cell by dividing the integrated capacitive transients by the voltage. Consistent with the short time in culture and lack of extensive processes, the capacitive transients of recorded cells had single-exponential decay phases. Input resistances were calculated from the steady-state currents elicited by the same ±10 mV steps (from the holding potential). For each cell, the series resistance was calculated by dividing the time constant of the decay of the capacitive transient (fit by a single exponential) by the $C_m$. Series resistances were routinely compensated ≥80%. Voltage errors resulting from the uncompensated series resistance were always ≤4 mV and were not corrected. Only data obtained from cells with input resistances >50 MΩ and capacitive transients well described by single exponentials were included in the cumulative data analyses presented.

## Measurements of $I_{NaT}$, $I_{NaR}$, and $I_{NaP}$

In the standard voltage-clamp studies, involving step depolarizations, peak transient Nav conductances (in each cell at each test potential) were calculated (using $E_{reversal}$ = +75 mV) from measurements of peak $I_{NaT}$ following digital subtraction of the non-inactivating, persistent Nav current component ($I_{NaP}$), measured during the same voltage step. These peak transient Nav conductances were then normalized to the maximal peak transient Nav conductance determined in the same cell. Mean ± SEM normalized peak transient Nav conductances were then determined, plotted as a function of the test potential, and fitted with a single (*Equation 1*) Boltzmann function:

$$G_{Na}/G_{NA,max} = 1/(1 + e[(V_h - V_m)k]) \tag{1}$$

where $V_h$ is the membrane potential of half-maximal activation and k is the slope factor.

To determine the voltage dependence of steady-state inactivation of $I_{NaT}$, a two-step voltage-clamp protocol was used. From a holding potential of −70 mV, brief (20 ms) voltage steps to various conditioning voltages, between −120 and −35 mV, were presented prior to depolarizations to 0 mV to evoke $I_{NaT}$. Peak $I_{NaT}$ amplitudes at 0 mV, evoked from each conditioning potential in each cell, were measured following digital subtraction of $I_{NaP}$, measured during the same voltage step, and subsequently normalized to the peak $I_{NaT}$ amplitude evoked (at 0 mV) from the −120 mV conditioning step in the same cell. Mean ± SEM normalized peak $I_{NaT}$ amplitudes were plotted as a function of the conditioning voltage and fitted with a single (*Equation 2*) Boltzmann function:

$$I_{Na}/I_{Na,max} = 1/(1 + e[(V_h - V_m)/k]) \tag{2}$$

where $V_h$ is the membrane potential of half-maximal inactivation and k is the slope factor.

The peak amplitudes of $I_{NaR}$, evoked on hyperpolarizations to various membrane potentials, following brief (5 ms) depolarizing voltage steps to 0 mV, were determined following digital subtraction of $I_{NaP}$, measured during the same voltage step (in the same cell). The time constants (tau) of the decay of $I_{NaR}$ were determined from first-order exponential fits to the decay phases of these subtracted records. The amplitudes of $I_{NaP}$ were routinely determined by direct measurement of the steady-state inward currents remaining at the end of depolarizing (or hyperpolarizing) voltage steps. $I_{NaP}$ (or $I_{NaP}$ plus $I_{NaR}$) amplitudes were also determined in additional experiments from analyses of the currents recorded during depolarizing (or hyperpolarizing) voltage ramps (see *Figure 6*).

The peak amplitudes of $I_{NaT}$ and $I_{NaR}$, generated using our Nav channel gating model or the Raman-Bean model, were routinely determined directly from the simulated current records, that is, without subtraction of $I_{NaP}$. Determinations of $I_{NaT}$ and $I_{NaR}$ amplitudes following subtraction of $I_{NaP}$ yielded

indistinguishable results. The time constants of decay of $I_{NaR}$ simulated using our model or the Raman-Bean model were determined from single exponential fits to the inactivating component of the current (i.e., $I_{NaR}$ uncontaminated by $I_{NaP}$).

Sample sizes were determined based on mean data (and associated standard deviations) obtained from analyses of voltage-clamp recordings of Nav currents obtained from mouse cerebellar Purkinje neurons (*Ransdell et al., 2017*). As in this previous study, all of the mean data presented here reflect results obtained from biological replicates, that is, analyses of Nav current recordings, evoked using identical voltage-clamp protocols, obtained from individual isolated Purkinje neurons. Technical replicates, that is, the presentation of identical voltage-clamp protocols to the same Purkinje neuron, were occasionally acquired to ensure that the properties of the Nav currents did not change during prolonged whole-cell recordings. Technical replicates were not included in the data analysis or reported results.

Data were analyzed using ClampFit (Molecular Devices), MATLAB (Mathworks), Microsoft Excel, and Prism (GraphPad Software Inc).

## Simulations using the Raman-Bean model

The Raman-Bean model of Nav channel gating (*Raman and Bean, 2001*) was coded in MATLAB using the equations and schematic of the Markov model structure presented in Figure 7 of *Raman and Bean, 2001*; no changes were made to the published model or parameters (*Raman and Bean, 2001*). The model simulations were run as described below using the matrix exponential technique (*Teed and Silva, 2016*).

## Development of a computational model of Nav channel gating

A Markov kinetic state model of Nav channel gating in mouse cerebellar Purkinje neurons was formulated based on our acquired experimental data that led to a hypothesized structure of channel gating states (see: *Figure 3A*). The rate constants in the model were all single exponentials derived from the acquired experimental data and were optimized numerically using described methods (*Moreno et al., 2016*; *Teed and Silva, 2016*). All simulations, numerical optimization, and data visualization were done in MATLAB 2017B. Detailed methods are below. Model definition files and MATLAB scripts used for the simulations are available at https://github.com/morenomdphd/Resurgent_INa; *Moreno, 2022*. The equations were verified by one of the authors (DB) who was not involved in the creation of the model.

The Nav channel gating model developed (see *Figure 3*) consists of nine states: three closed states (C1, C2, C3), a single open state (O), two closed-inactivated states (IC1, IC2), two inactivated states, resulting from fast channel inactivation (IF1, IF2), and an additional, slowly populated inactivated state (IS). The model accurately simulates the experimentally observed kinetics and voltage dependences of Nav channel gating, including activation, inactivation (closed and open state), deactivation, recovery from fast inactivation, as well as the voltage dependence of activation and the proportion (relative to $I_{NaT}$) of $I_{NaR}$.

## Numerical optimization procedure

All computations were done in MATLAB 2017B. We coded all voltage protocols and used the matrix exponential technique, which is described in *Teed and Silva, 2016*, for simulations. A modified Nelder-Mead simplex method that allows for constrained optimization (only positive rate constants) was used for simultaneous optimization of the protocols listed below. A cost function for each protocol was defined as the sum of squared differences between the experiments and the simulations. The total cost function (sum of the individual protocol errors) was then minimized and converged when a tolerance of 0.01 for the change of the cost function and 0.01 for the change in parameters was achieved. For further details about this numerical optimization method, see *Moreno et al., 2016*.

The protocols were as follows:

- Steady-state availability: For each voltage between –120 and –10 mV, the steady-state probabilities of the channel were found. The channel was then depolarized to 0 mV, and the open-state probability was determined. The value of the open-state probability was then normalized to the open-state probability at –120 mV.

- Steady-state activation: Channel steady state was found at –80 mV. The channel was then depolarized to voltages between –77 and 0 mV (in 3 mV increments). For each voltage, the maximum open probability of the channel was calculated, and the conductance, $G_{Na}$, at each voltage was determined. The calculated values were then normalized to $G_{Na}$ at 0 mV.
- Tau of inactivation: Channel steady state was found at a holding potential of –90 mV. From steady state, the channel was depolarized to voltages between –50 and 0 mV (in 5 mV increment), and the time constant (tau) corresponding to 64% decay (1/e) of the peak current was calculated for each voltage.
- Recovery from fast inactivation: From a steady state of –90 mV, the channel was depolarized to 0 mV, and the peak current determined. The channel was then allowed to recover at –90 mV for variable time intervals, before being depolarized again to 0 mV. The peak currents during the depolarizations to 0 mV following the various recovery times were determined and normalized to the initial peak current.
- Persistent component of the Nav current: From a steady state of –90 mV, the channel was depolarized to 0 mV for 5 ms, and subsequently hyperpolarized to –45 mV for 100 ms. The persistent current recorded after the 100 ms hyperpolarizing step was measured and normalized to the initial peak current.
- Voltage dependence of the ratio of the peak resurgent to peak transient Nav current amplitude: From a steady state of –90 mV, the channel was depolarized to 0 mV for 5 ms, and subsequently hyperpolarized to potentials between –5 and –80 mV (in 5 mV increments). The peak resurgent current at each hyperpolarized voltage step was determined and normalized to the peak transient inward current evoked at 0 mV.
- Dependence of the peak $I_{NaR}$ amplitude on the duration of the depolarizing voltage step duration: From a steady state of –90 mV, the channel was depolarized to +20 mV for varying times (2 to 36 ms) prior to hyperpolarization to –45 mV. The amplitude of $I_{NaR}$ at –45 mV following each depolarizing voltage step to +20 mV (of varying durations) was measured and normalized to the peak $I_{NaR}$ evoked following the 2 ms depolarizing voltage step.
- Tau of decay of the resurgent Nav current: From a steady state of –90 mV, the channel was depolarized to 0 mV for 5 ms, and subsequently hyperpolarized to potentials ranging from –5 to –45 mV (in 5 mV increments). The time constant (tau) of the exponential decay (1/e) of the resurgent current at each hyperpolarized test potential was determined.
- Voltage independence of peak resurgent current: Consistent with the experimental data presented in *Figure 2D and E*, the peak resurgent current was independent of the potential of the depolarizing voltage step. From a steady state of –90 mV, the channel was depolarized to membrane potentials ranging from –5 to –35 mV (in 5 mV increments) prior to a hyperpolarizing voltage step to –45 mV. The difference between the peak $I_{NaR}$ measured at –45 mV from each depolarized potential was determined, and the mean resurgent peak current determined across all depolarized potentials was minimized. This ensured the peak resurgent current was constant without specifying the magnitude of the resurgent peak a priori.

## Optimization of the Scn4b[-/-] Nav current model

To develop the model for Nav channels lacking *Scn4b*, the optimization routine was restarted from the optimized wild type rate constants (initial conditions). The 'Voltage dependence of the ratio of the peak resurgent to peak transient Nav current amplitude' protocol described above was fitted to the experimental data acquired from isolated cerebellar Purkinje neurons from (*Scn4b[-/-]*) animals harboring a targeted disruption in the *Scn4b* locus (*Ransdell et al., 2017*); the data are presented in *Figure 9A*. All of the other protocols used in the optimization procedure for *Scn4b[-/-]* were the same as those described above for the wild type Nav channel to ensure no other changes to the model.

## Model parameters

| Rate parameter | Optimized rate (WT) | Optimized rate (Scn4b$^{-/-}$) | Rate equations |
|---|---|---|---|
| a11_variable1 | 2.3989e-02; | 2.0761e-02; | |
| a11_variable2 | 9.6108e + 02; | 9.7685e + 02; | |
| a12 | 8.5613e + 02; | 8.3340e + 02; | |
| a13 | 7.2682e + 01; | 6.1087e + 01; | |
| b11_variable1 | 1.7233e-01; | 1.6995e-01; | |
| b11_variable2 | 1.9691e + 01; | 1.8560e + 01; | |
| b12 | 8.8549e + 01; | 9.3042e + 01; | |
| b13 | 1.4841e + 02; | 1.7914e + 02; | Where the transition rates are of the form: a11 = Tfactor*1/(Input(1)*exp(-V/Input(2))); a12 = |
| a3_variable1 | 3.6734e-01; | 3.9273e-01; | Input(3)*a11; a13= Input(4)*a11; b11 = Tfactor*1/ |
| a3_variable2 | 9.8034e + 02; | 9.8807e + 02; | (Input(5)*exp(V/Input(6))); b12 = Input(7)*b11; |
| b3_variable1 | 5.3241e + 01; | 4.7402e + 01; | b13= Input(8)*b11; a3 = Tfactor*Input(9)*exp(-V/ |
| b3_variable2 | 1.4204e + 01; | 1.3469e + 01; | Input(10)); b3 = Tfactor*Input(11)*exp((V)/Input(12)); |
| a2_variable1 | 8.7852e + 01; | 8.1085e + 01; | a2 = Tfactor*(Input(13)*exp(V/Input(14))); b2 = |
| a2_variable2 | 9.9972e + 02; | 9.9984e + 02; | ((a13*a2*a3)/(b13*b3)); a6 = Tfactor*(Input(15)*exp(V/ |
| a6_variable1 | 6.0921e + 02; | 6.0756e + 02; | Input(16))); b6 = Tfactor*Input(17)*exp(-V/Input(18)); |
| a6_variable2 | 8.6490e + 01; | 7.8591e + 01; | a2s = Tfactor*(Input(19)*exp(V/Input(20))); b2s |
| b6_variable1 | 1.5645e + 02; | 1.6214e + 02; | = Tfactor*(Input(21)*exp(-V/Input(22))); a3s = |
| b6_variable2 | 3.0317e + 01; | 4.3656e + 01; | Tfactor*Input(23)*exp(-V/Input(24)); b3s = (a2s*a3s*a13)/ |
| a2s_variable1 | 1.5817e + 01; | 2.3130e + 01; | (b2s*b13); |
| a2s_variable2 | 9.9982e + 02; | 9.9969e + 02; | Q10 = 3; T = 295; |
| b2s_variable1 | 1.0010e-03; | 1.0208e-03; | |
| b2s_variable2 | 1.1963e + 01; | 1.1383e + 01; | Tfactor = 1.0/(Q10^((37.0-(T-273))/10.0)); |
| a3s_variable1 | 4.4773e-03; | 2.8677e-03; | Note: **b2, b3s** are constrained by microscopic |
| a3s_variable2 | 9.9993e + 02; | 9.8330e + 02; | reversibility. |

Of note, in the MATLAB script, the Nav channel model developed here contains 24 parameters; these are inputted as a matrix 'Input'. For example, Input(1) corresponds to a11_variable1, and Input(2) corresponds to a11_variable2, Input(3) corresponds to a12, and Input(4) corresponds to a13, etc. The transition rate constants are of the form denoted in the right-hand column of the table above.

## Acknowledgements

The authors thank Mr Richard Wilson for expert technical assistance. The financial support provided by the NIH (R01 NS065761 to JMN, R01 HL136553 to JRS, and F32 NS090765 to JLR) is also gratefully acknowledged; JDM was supported by an NIH institutional training grant (T32 HL007081) and a grant from the Foundation for Barnes Jewish Hospital. The authors declare no competing financial interests.

## Additional information

### Funding

| Funder | Grant reference number | Author |
|---|---|---|
| National Institute of Neurological Disorders and Stroke | NS065761 | Jeanne M Nerbonne |
| National Heart, Lung, and Blood Institute | HL136553 | Jonathan R Silva |
| National Institute of Neurological Disorders and Stroke | NS090765 | Joseph L Ransdell |

The funders had no role in study design, data collection and interpretation, or the decision to submit the work for publication.

### Author contributions

Joseph L Ransdell, Jonathan D Moreno, Conceptualization, Data curation, Formal analysis, Investigation, Methodology, Project administration, Writing – original draft, Writing – review and editing; Druv Bhagavan, Jonathan R Silva, Conceptualization, Data curation, Formal analysis, Funding acquisition,

Investigation, Methodology, Project administration, Writing – review and editing; Jeanne M Nerbonne, Conceptualization, Funding acquisition, Investigation, Methodology, Project administration, Writing – review and editing

## Author ORCIDs
Joseph L Ransdell ![ORCID] http://orcid.org/0000-0001-5908-9044
Jonathan R Silva ![ORCID] http://orcid.org/0000-0002-3696-3955
Jeanne M Nerbonne ![ORCID] http://orcid.org/0000-0001-8334-8499

## Ethics
This study was performed in strict accordance with the recommendations in the Guide for the Care and Use of Laboratory Animals of the National Institutes of Health. All of the animals were handled according to an approved institutional animal care and use committee (IACUC) protocol (20180045) of Washington University in St. Louis (Animal Welfare Assurance # A-3381-01).

## Decision letter and Author response
Decision letter https://doi.org/10.7554/eLife.70173.sa1
Author response https://doi.org/10.7554/eLife.70173.sa2

---

# Additional files

## Supplementary files
• Transparent reporting form

## Data availability
Model definition files and Matlab scripts used for the simulations are available at https://github.com/morenomdphd/Resurgent_INa, (copy archived at swh:1:rev:07202f3d8c299b3b918fd9ae91b005a07d34c095).

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
