## [Editor Report]

After more than 20 years of intensive research the molecular machinery of Resurgent Currents (INaR), a non-canonical identity of currents mediated by voltage-activated sodium channels is still a mystery. In this paper, Ransdell and colleagues advance the conceptual framework with new experimental insight and a new kinetic model of INaR.

---

## [Decision Letter]

**Decision letter after peer review:**

Thank you for submitting your article "Intrinsic Mechanisms in the Gating of Resurgent Na^+^ Currents" for consideration by *eLife*. Your article has been reviewed by 3 peer reviewers, including Teresa Giraldez as Reviewing Editor and Reviewer #1, and the evaluation has been overseen by Richard Aldrich as the Senior Editor.

Essential revisions:

All reviewers agreed that these results are interesting and compelling, offering a fresh and stimulating perspective and including some significant new experimental data. However, reviewers also raised several concerns about the interpretation of the data and some other aspects related to implementation of the model that should be addressed by the authors. Essential revisions should include:

1) Address the possible limitations of the new model according to the comments of the reviewers.

2) Discussion of whether the model can predict the different kinetics of recovery from inactivation of INaT and INaR.

3) Some discussion of whether the different inactivated states can be related to the molecular machinery of the channel as it is currently known.

4) Reconcile discrepancies and apparent inconsistencies in the experimental data. The authors should at least address the apparent inconsistency between their data and: a) previous experimental results for the size of the resurgent current relative to transient current, and b) discuss how their experimental data with native resurgent current are similar or different from previous results using free charged peptide pieces from the beta4 subunit (where an open channel block mechanism seems likely).

5) Discussion about interpretation of Figure 6.

Because the individual reviews include several important points they are included here for you reference.

*Reviewer #1 (Recommendations for the authors):*

According to the open block model (Raman and Bean, 2001), the INaP current component represents the dynamic distribution of sodium channels between open blocked, open and normal inactivated states. How could the new Markov model describe the possible gating transitions involved in the generation of INaP? Related to this, the statement in lines 506 may confuse some readers, is it specifically shown in this manuscript that the amplitude and voltage-dependence of INaR mirrors that of INaP?

In Figure 2A, INaP is not clearly seen as in e.g. Figure 1A, and even appears to be slightly outwards, which would not be consistent with a Na^+^ current if ENa is +75 mV (as stated in the text). Please explain. Was there a high variability in INaP between cells? This may be relevant, considering that the relative amplitude of INaP is critical in determining the magnitude of INaR (as the authors state in the discussion).

What is the prediction of the open block model on INaR gating at different membrane depolarizations? Adding this Figure (as supplementary) would compare and contrast all of your experimental findings with both models.

The experimental data in Figure 8 reveals that the INaR channels are populated separately with slow inactivated state and the novel gating model recapitulates the data credibly. Conversely the open channel block model fails to simulate these results (Supp Figure 2A). Can the OB model simulate any of the experimental data from Scn4b-/- neurons? It would be interesting to see the relative occupancy plots with this model.

The simulation data in Figure 3H Vs. Figure 4B is different. In Figure 4B (bottom panel), when the depolarization at 0 mV is longer, the activation time of INaR is slower and the decay time of both INaT and INaR are slower. This seems not consistent with Figure 3H. Please explain.

*Reviewer #2 (Recommendations for the authors):*

1) Generally the results reported for the Raman-Bean model seemed correct. However, one did not: the plot of decay time constants in Figure 7H. The Raman-Bean paper shows a decay time constant of 25 msec at -30 mV (their Figure 8B), while this plot gives its value as about 45 ms. And the discrepancy seems even greater at -60 mV, where the modeled results in Figure 8A of the Raman-Bean seem to have a time constant of maybe 5 ms, while in Figure 7 H the values in this region are far higher. The authors should check this point.

2) A major experimental difference between the results here and previous reports is the magnitude of resurgent current relative to transient current. For example, the Raman- Bean gave average values of 6.8 nA for transient current and 193 pA for maximal resurgent current, so resurgent current was ~3% of transient current. Similar values are evident in most of the previous papers from the Raman lab, e.g. the most recent paper by White et al., (2019) gave a mean value of 2.7% for Purkinje neurons from wild-type mice. In contrast, here the authors give experimental values of ~27% (Figure 7F). This seems very surprising and is obviously an important point because it is one of the biggest differences in the predictions of the two models. The authors should at least comment on this difference in the experimental data.

3) The data in Figure 6 is interpreted as showing that the voltage-dependence of persistent and resurgent sodium current are identical. It is not obvious to me that the current during the “reverse” voltage ramp is primarily resurgent current rather than persistent current. In principle, from the authors’ earlier results it would be expected that resurgent current would be substantially inactivated by the relatively long time spent at depolarized voltages before the ramp down reaches ~-30 mV where the current is biggest. In contrast, persistent sodium current would not be expected to inactivate. What is the argument that the current during the reverse ramp is mostly resurgent current and not simply mostly persistent current? It is not obvious to me that this experiment would look much different in a cell type that did not have resurgent sodium current.

*Reviewer #3 (Recommendations for the authors):*

Dependency of InaR on depolarizing potential and time

In previous experiments, InaR was slightly dependent on the magnitude of the depolarization step, especially at higher potentials (here it was only tested up to +10mV) (Biophys J. 2001 Feb;80(2):729-37., also own results with β 4 petide). Also if the depolarization step is prolonged, InaR becomes strongly dependent on voltage (Biophys J. 2007 Mar 15;92(6):1938-51, Figure 5B, J Neurosci. 2010 Apr 21;30(16):5629-34.) Therefore, the claims of the authors need more experimental backup.

[Editors’ note: further revisions were suggested prior to acceptance, as described below.]

Thank you for resubmitting your work entitled “Intrinsic Mechanisms in the Gating of Resurgent Na^+^ Currents” for further consideration by *eLife*. Your revised article has been evaluated by Richard Aldrich (Senior Editor), a Reviewing Editor, and the original reviewers.

After the consultation session, all reviewers agree that the authors’ main finding that the resurgent current can be modelled by a mechanism independent of open-channel block is very relevant and would certainly encourage the field with fresh ideas. The manuscript has been improved after revision. However, two reviewers still found remaining issues that need to be addressed, as outlined below:

1) Some ambiguity exists about how “InaT”, “InaR”, and “InaP” are defined and measured in both the data and the results of the models. The authors should include a section in the Methods to clearly state how the components of current termed “InaT”, InaR”, and “InaP” were defined for both experimental measurements and modelling, and specifically whether quantification of the size of the “InaR” current seen upon hyperpolarization following depolarizations was only the decaying current or included the steady-state current that could be considered “InaP”.

2) Optionally, the authors could consider revising the manuscript according to the second reviews listed below.

*Reviewer #1 (Recommendations for the authors):*

The authors have addressed all the concerns. I have no further comments.

*Reviewer #2 (Recommendations for the authors):*

The authors have responded at length in the “Response to Reviews” document, although in many cases, the actual changes in the manuscript are fairly minimal. In my opinion, several of the points that were raised in the initial review remain unclear or confusing in the presentation of the manuscript, as discussed below. It seems that several of these reflect an ambiguity about how (or even whether) measurements of “InaR” distinguish between resurgent sodium current and persistent sodium current.

1) A point that was raised in in the previous reviews by both Reviewer 1 and Reviewer 2 is the interpretation of the experiment in Figure 6 (similar voltage-dependence of currents evoked by slow ramps either in the depolarizing or hyperpolarizing direction) as evidence that the voltage-dependence of “InaR” and “InaP” are the same. In response to the point raised by Reviewer 2 that the hyperpolarizing ramps may be almost all persistent sodium current because the resurgent current may be mostly inactivated before or during the hyperpolarizing ramp, the authors acknowledge this “excellent” point, but this isn’t reflected in the actual manuscript. They have simply added lines 242-242 giving the original interpretation (that depolarizing ramps evoke only InaP and hyperpolarizing ramps InaP plus InaR), with no discussion of the size of the “extra” InaR relative to “InaP or the fact that “InaR” might be mostly inactivated. In fact, the current evoked by the hyperpolarizing ramp is the same magnitude as the depolarizing ramp, which seems most consistent with the idea that resurgent current has been totally inactivated before the hyperpolarizing ramps begins. In that case, ramps in both directions simply measure steady-state persistent current, and there is no justification for using this experimental result as evidence that the voltage-dependence of the resurgent current is the same as persistent current.

2) A point that may be related: in both the original manuscript and the revised manuscript, it seems that often in referring to measurements of “InaR” – both for experiments and models – the authors are not distinguishing between components of resurgent current and persistent current, and the measurement of “InaR” refers to the combination of both. This is never really made clear in the manuscript, either in Methods or Results. The fact that they may consider “InaR” to refer both to decaying and steady-state components of current is suggested by the author’s response to the issue of the time constant for decay of resurgent current predicted by the Raman-Bean 2001 model. The authors argue that their value of a time constant of 45 ms for the decay of resurgent current at -30 mV predicted by this model was not a mistake, but rather some sort of error in the 2001 paper in reporting a value of 25 ms for the Raman-Bean model (“We are not certain how the value of 25.4 ms reported in Raman and Bean, 2001 was determined.”). In fact, it seems very clear that the difference is that the exponential fit in the Raman-Bean paper is only the decaying part of the current, i.e. decay of the resurgent current to a non-zero steady-state value that reflects InaP. This is clearly stated in that paper, and is clearly illustrated in the fits shown for both the experimental data (Figure 1B) and the results of the model (Figure 8B). To check if the authors’ value of 45 ms could reflect some discrepancy in the actual odelled here and in the 20021 Raman-Bean paper, I digitized the trace the authors include in the Response to Reviews showing their calculation for resurgent current at -30 mV predicted by Raman-Bean model. With crude digitization of the trace, the trace is fit by a time constant is 27 ms, not 45 ms – if the fit allows for the steady-state value of non-decaying current. The authors do not show a fit to their trace, and the trace is truncated before the steady-state is clearly shown, but there is no way the trace is well-fit by an exponential with a time constant of 45 ms. The only way I can see of coming up with a time-constant of 45 msec would be to force a fit that decays to zero, which is clearly inappropriate for a current that decays to a steady-state.

These are details that do not detract from the main point of the manuscript that resurgent current can be interpreted without invoking an open-channel blocking mechanism. However, readers will benefit by a clear statement about whether and how components of NaR and InaP are distinguished in the various measurements that are reported, both for the experimental data and for both models that are discussed.

*Reviewer #3 (Recommendations for the authors):*

The authors addressed all the questions raised by the reviewers in considerable detail. However, they did not touch the model nor did they revise their interpretation of the model. Therefore, my issues previously raised remain at the core. Below I have tried to clarify my concerns. Taken together the new model is not so different from the Raman and Bean (RandB) model. While it fares better in some aspects, it introduces new issues, especially with InaT inactivation and persistent current. Furthermore, it might be that the RandB model could approximate the new data far better than anticipated here, if only optimized (and maybe tested with a second IF =OB state). Finally, I do not agree with the interpretation that InaT and InaR are two completely distinct identities in their model as outlined below.

InaT and InaR decay:

The authors did not follow the implications of my comment. Because of the single open state InaT and InaR must! Decay with the same kinetics. Apparently, they do as shown in Figure 2D at -45mV. The different decay is only due to using different voltages for comparison and of course different behavior of the IF/IS states at different voltages. Therefore InaT and InaR mechanisms are only distinct because of membrane potential (or its history) and not per se. The wording throughout the results part is still not reflecting this matter.

In addition, in Figure 4B during bulk InaT IS, IF1 and IF2 become all populated quiet fast indicating IS is not a slow inactivated state, just a bit slower at depolarized potentials compared to IF. If stated this way, the new model is not very different from the RandB model. My interpretation of the new model is that at high potentials IF states adsorb fast and dissipate slowly (Figure 4B) while at lower potentials the equilibrium changes and they dissipate fast and do not adsorb (Figure 4B). Note, this is not different from the OB state in the RandB model (despite two IF states here). Second, the IS state is always absorbing and becomes dominant at more negative potentials. Therefore the IS state is not qualitatively different from the inactivated states in the Raman and Bean model. Note how similar the occupancies are in Figure 4A+B (->IS) vs. Figure 7B (->I). However, the new model differs in two important aspects, a second IF state and a shortcut from IF to C. Are these additions required? See my comment below.

Fast inactivation:

The many inactivation time constants result in a strange looking slow decay component, not compatible with real data. This is becoming obvious in Figure 3H,I green traces. That is a kink in InaT and a slow decrease of InaP presumably due to slow accumulation in IS. Pointing out that three time constants of inactivation (+ transition to C) are likely too much.

The fit of real data shown for reference in the response of the authors is not convincing, also because all components would not be covered with a single-exponential fit.

Kinetic scheme:

I do not agree with the authors here on one point important for the proposed model. In the Raman-Bean model, the blocked state could principally slowly dissipate through an increased persistent current, which is indeed visible in Fig, 7H. If the new data (with or without adding a second IF state to OB) is fitted to the RandB model, is the modified model then capable of fitting the decreased InaR and the data from Figure 8 and therefore not inferior to the proposed model here?

Differential recovery of InaT and InaR:

I absolutely agree that two populations of sodium channels could be the cause as pointed out in the discussion. Given that, the arguments brought forward (line 214-233) are becoming void.

---

## [Author Response]

Essential revisions:All reviewers agreed that these results are interesting and compelling, offering a fresh and stimulating perspective and including some significant new experimental data. However, reviewers also raised several concerns about the interpretation of the data and some other aspects related to implementation of the model that should be addressed by the authors. Essential revisions should include:1) Address the possible limitations of the new model according to the comments of the reviewers.

We have addressed each of the Reviewer’s specific comments in the paragraphs that follow and we have revised the text to address the limitations of the presented Markov model (lines 429-435 and 470-478).

2) Discussion of whether the model can predict the different kinetics of recovery from inactivation of InaT and InaR.

We have responded to the question (from Reviewer three) about whether the model can reproduce the differential recovery from inactivation (observed experimentally) of I_NaT_ and I_NaR_ in our response to Reviewer three below. We have also revised the manuscript to address this point directly (lines 470-478).

3) Some discussion of whether the different inactivated states can be related to the molecular machinery of the channel as it is currently known.

We discuss our current understanding of the molecular machinery underlying Nav channel gating, and how this understanding fits with proposed model for the gating of I_NaR_ in the revised manuscript (lines 449-478).

4) Reconcile discrepancies and apparent inconsistencies in the experimental data. The authors should at least address the apparent inconsistency between their data and: a) previous experimental results for the size of the resurgent current relative to transient current, and b) discuss how their experimental data with native resurgent current are similar or different from previous results using free charged peptide pieces from the beta4 subunit (where an open channel block mechanism seems likely).

We have responded to the Reviewer’s comments regarding differences in the measured amplitudes of I_NaR_ (relative to I_NaT_) and clarified the experimental conditions used to record I_NaR_ (lines 162-163 and 278-282). However, because previous studies conducted using the beta4 peptide to generate resurgent Nav currents were conducted in cell types other than cerebellar Purkinje neurons, we have not directly compared the properties of these currents to the properties of the native resurgent currents in Purkinje neurons detailed here.

5) Discussion about interpretation of Figure 6.

We have revised the manuscript (lines 242-244) to clarify how we interpret the evoked inward current measured during the repolarizing voltage ramp protocol (in Figure 6).

Because the individual reviews include several important points they are included here for you reference.Reviewer #1 (Recommendations for the authors):According to the open block model (Raman and Bean, 2001), the InaP current component represents the dynamic distribution of sodium channels between open blocked, open and normal inactivated states. How could the new Markov model describe the possible gating transitions involved in the generation of InaP? Related to this, the statement in lines 506 may confuse some readers, is it specifically shown in this manuscript that the amplitude and voltage-dependence of InaR mirrors that of InaP?

Thank you for this comment. Similar to the Raman and Bean (2001) model, the novel model presented produces a persistent sodium current component, reflecting the portion of channels that are not accumulated in the fast-inactivated state during membrane depolarizations. This is accomplished through the dynamic distribution of channels between the open (conducting) state and the fast- and slow-inactivating states.

The statement (on line 506 of the original manuscript) was in reference to the experimental result presented in Figure 6. In the revised manuscript, we made this reference to the experimental data clear (lines 354-356). We have also removed the mention of I_NaR_ amplitude from this discussion point.

In Figure 2A, InaP is not clearly seen as in e.g. Figure 1A, and even appears to be slightly outwards, which would not be consistent with a Na^+^ current if Ena is +75 mV (as stated in the text). Please explain. Was there a high variability in InaP between cells? This may be relevant, considering that the relative amplitude of InaP is critical in determining the magnitude of InaR (as the authors state in the discussion).

We agree that I_NaP_ is larger (and, therefore, clearer) in the record presented in Figure 1A, compared with the record shown in Figure 2A. In general, I_NaP_ is very small, compared with I_NaT_ and I_NaR_ and, therefore, difficult to measure at depolarized potentials. It is also correct, as this Reviewer suggests, that I_NaP_ is variable across cells. It is certainly also possible there is a small contaminating outward current in the recording illustrated in Figure 2A. In our experience, reliable measurement of I_NaP_ requires the use of tetrodotoxin (TTX) to isolate the TTX-sensitive currents, which was not done here. It will be interesting to quantify cell-cell variability in the magnitude of I_NaP_ and the relative I_NaP_ to I_NaR_ amplitudes across cells; these analyses, however, will need to be conducted on TTX-subtracted currents.

What is the prediction of the open block model on InaR gating at different membrane depolarizations? Adding this Figure (as supplementary) would compare and contrast all of your experimental findings with both models.

Thank you for this interesting question. To respond, we have performed simulated voltage-clamp experiments using the open block model and present the resulting traces below. As is evident, the open block model produced I_NaR_ records that are not critically dependent on the voltage of the initial depolarization. Although these findings were initially surprising, further analysis revealed that this is due to the rate constants (used in the open block model) of the O -> OB transition are voltage independent, while the rate constants of the OB -> O transition are voltage dependent. This aspect of the model is surprising given that one important premise of the open-channel block hypothesis is that Nav channels are favored to enter into the open-blocked state at more positive depolarizations and favored to enter into the conventional inactivated state at moderate depolarizations.

**Author response image 1. sa2fig1:** Simulations of I_NaR_ following membrane depolarizations to different voltages using the open-blocked model. *Left:* voltage protocol indicating a 0 mV (black), -15 mV (blue), -30 mV (red), -45 mV (green) depolarization for 5 ms from a membrane potential of -90 mV. *Middle*: Simulated Nav current traces. *Right:* A zoomed in view of the simulated Nav currents reveals that the open-blocked model produces resurgent currents quite different from those observed experimentally (Figure 2A); compare with the results of the novel model generated here (Figure 3I). Additionally, I_NaR_ inactivation in this (open-blocked) model appears to show some voltage dependence, which is also not observed in our model (Figure 3I).

The experimental data in Figure 8 reveals that the InaR channels are populated separately with slow inactivated state and the novel gating model recapitulates the data credibly. Conversely the open channel block model fails to simulate these results (Supp Figure 2A). Can the OB model simulate any of the experimental data from Scn4b-/- neurons? It would be interesting to see the relative occupancy plots with this model.

This is an interesting question. The OB model is expected to fail to simulate the *Scn4b^-/-^* model in the same fashion as the WT model for the multi-pulse plot noted in Figure 8-supplement 1A. This is because the only substantive difference between the WT and *Scn4b^-/-^* datasets is the magnitude of the resurgent current; the kinetics and voltage-dependences of the currents are very similar (by design) and, as a result, the *Scn4b^-/-^* model fits the data fairly well.

We have previously reported that peak I_NaR_ in *Scn4b^-/-^* Purkinje neurons was reduced compared to wild type controls, whereas we found no differences in I_NaR_ kinetics or voltage-dependences in wild type and *Scn4b^-/-^* Purkinje neurons (Ransdell et al., 2017); this is now noted on lines 116-118 and 324-326. We did not perform experiments on *Scn4b^-/-^* Purkinje neurons using the voltage-clamp protocol in Figure 8.

The simulation data in Figure 3H Vs. Figure 4B is different. In Figure 4B (bottom panel), when the depolarization at 0 mV is longer, the activation time of InaR is slower and the decay time of both InaT and InaR are slower. This seems not consistent with Figure 3H. Please explain.

We apologize for the confusion. The time base in Figure 4B (a “zoomed in” version of 3H) is very different from the time base in Figure 3H; note the differences in the scale bars (2 ms versus 20 ms).

Reviewer #2 (Recommendations for the authors):1) Generally the results reported for the Raman-Bean model seemed correct. However, one did not: the plot of decay time constants in Figure 7H. The Raman-Bean paper shows a decay time constant of 25 msec at -30 mV (their Figure 8B), while this plot gives its value as about 45 ms. And the discrepancy seems even greater at -60 mV, where the odelled results in Figure 8A of the Raman-Bean seem to have a time constant of maybe 5 ms, while in Figure 7 H the values in this region are far higher. The authors should check this point.

Thank you for identifying this confusing point. Please allow us to clarify. The plots in 7H (Raman-Bean model) and 3F (our model) are not the decay times of the resurgent currents. Rather, these are plots of the normalized peak resurgent currents as a function of the duration of the depolarizing step (prepulse) to +20mV. Thus, as one maintains the depolarizing prepulse to +20 mV for 40 ms, our experimental data (plotted as points in Figures 3F and 7H) reveal that the peak resurgent current, when normalized to the current observed after a 2 ms depolarizing step (to +20mV), decays to ~50% of its maximal value and, in addition, that this result is reproduced in our model (solid line in Figure 3F). In contrast, the Raman-Bean model predicts that there is very little decay of the resurgent current as a function of duration of the depolarizing prepulse (solid line in Figure 7H).

We also considered that this Reviewer might be referring to Figure 7I in which the I_NaR_ decay time constants are plotted as a function of voltage. We double checked the accuracy of this plot and confirmed that the Raman-Bean model is as published in Raman and Bean, 2001. We have replotted Figures 8A and B from that paper, see Author response image 2. As shown in the inserts on the right sides of both panels, we have reproduced the results shown in the original Figures 8A and 8B of Raman and Bean, 2001. The tau of I_NaR_ decay (at -30 mV), determined from a single exponential fit to the decay phase of the simulated current, was ~45 ms, in agreement with the results presented in Figure 7I. We are not certain how the value of 25.4 ms reported in Raman and Bean, 2001 was determined.

2) A major experimental difference between the results here and previous reports is the magnitude of resurgent current relative to transient current. For example, the Raman- Bean gave average values of 6.8 nA for transient current and 193 pA for maximal resurgent current, so resurgent current was ~3% of transient current. Similar values are evident in most of the previous papers from the Raman lab, e.g. the most recent paper by White et al., (2019) gave a mean value of 2.7% for Purkinje neurons from wild-type mice. In contrast, here the authors give experimental values of ~27% (Figure 7F). This seems very surprising and is obviously an important point because it is one of the biggest differences in the predictions of the two models. The authors should at least comment on this difference in the experimental data.

Thank you for bringing up this important difference. In the White et al., 2019, as well as in the 1997 and 2001 Raman and Bean manuscripts, the extracellular ACSF used had a 50 mM Na^+^ concentration. In our studies, however, we routinely recorded I_NaR_ using ACSF with physiological (151 mM) Na^+.^ The higher Na^+^ concentration results in larger I_NaR_. In addition, there may be space-clamp issues that limit the ability to reliably measure the peak amplitudes of I_NaT_ evoked with physiological extracellular Na^+^; both of these technical issues could increase I_NaR_/I_NaT_ ratios. Consistent with this hypothesis, Levin and colleagues (2006) reported a I_NaR_/I_NaT_ ratio of 15 ± 2 %, i.e., a value substantially higher than the value of 3% reported previously by Raman and Bean (2001). We have now revised the text to highlight that we used a 151 mM external Na^+^ concentration (lines 526-527) in most experiments and to note that the Raman Bean model was developed from data derived from experiments using 50 mM NaCl (lines 278-282).

3) The data in Figure 6 is interpreted as showing that the voltage-dependence of persistent and resurgent sodium current are identical. It is not obvious to me that the current during the "reverse" voltage ramp is primarily resurgent current rather than persistent current. In principle, from the authors' earlier results it would be expected that resurgent current would be substantially inactivated by the relatively long time spent at depolarized voltages before the ramp down reaches ~-30 mV where the current is biggest. In contrast, persistent sodium current would not be expected to inactivate. What is the argument that the current during the reverse ramp is mostly resurgent current and not simply mostly persistent current? It is not obvious to me that this experiment would look much different in a cell type that did not have resurgent sodium current.

This is an excellent point. We agree the long duration of the ”reverse” voltage-ramp is likely to have resulted in inactivation of a substantial portion of I_NaR_ prior to the peak I_NaR_. Measuring the non-inactivating portion of the Nav current using ascending and descending voltage ramps, however, was difficult, and ramp protocols with higher dV/dt rates were not successful. We would also point out that the current amplitudes plotted in Figure 6B are similar throughout the voltage range. This is important because the membrane voltage is -10 mV very soon after the onset of the descending voltage ramp (when less inactivation would have taken place) and very late after the onset of the ascending voltage ramp. Additionally, this experiment shows that the non-inactivating Nav current can be evoked from a depolarized potential, consistent with recovery from inactivation into an open/conducting state. We have revised the manuscript (lines 242-244) to highlight these points.

Reviewer #3 (Recommendations for the authors):Dependency of INaR on depolarizing potential and timeIn previous experiments, INaR was slightly dependent on the magnitude of the depolarization step, especially at higher potentials (here it was only tested up to +10mV) (Biophys J. 2001 Feb;80(2):729-37., also own results with β 4 petide). Also if the depolarization step is prolonged, INaR becomes strongly dependent on voltage (Biophys J. 2007 Mar 15;92(6):1938-51, Figure 5B, J Neurosci. 2010 Apr 21;30(16):5629-34.) Therefore, the claims of the authors need more experimental backup.

We agree that a more systematic analysis of voltage- and time-dependence of I_NaR_ may be useful in further developing/improving the accuracy of the model. However, we do not think that these types of studies will help to determine if I_NaR_ is reflective of channels recovering from conventional (fast) inactivation or if the decay of I_NaR_ is reflective of Nav channels accumulating into a slow-inactivated state. Our plan is to proceed with experimental studies to directly test the hypothesis put forth in this study, and to use modeling to further refine our predictions.

[Editors' note: further revisions were suggested prior to acceptance, as described below.]

After the consultation session, all reviewers agree that the authors' main finding that the resurgent current can be modeled by a mechanism independent of open-channel block is very relevant and would certainly encourage the field with fresh ideas. The manuscript has been improved after revision. However, two reviewers still found remaining issues that need to be addressed, as outlined below:1) Some ambiguity exists about how "INaT", "INaR", and "INaP" are defined and measured in both the data and the results of the models. The authors should include a section in the Methods to clearly state how the components of current termed "INaT", INaR", and "INaP" were defined for both experimental measurements and modeling, and specifically whether quantification of the size of the "INaR" current seen upon hyperpolarization following depolarizations was only the decaying current or included the steady-state current that could be considered "INaP".

As requested, we have revised the Methods to include a section titled “Measurement of I_NaT_, I_NaR_, and I_NaP_” (lines 548-582). Additionally, we note that in revising the manuscript, we identified an error in the measurements of the rates of I_NaR_ decay determined for the currents using the Raman-Bean model (plotted in Figure 7I). We have corrected the error by recalculating the time constants of I_NaR_ decay and have replaced Figure 7I to reflect the corrected values. Finally, we have responded to the additional comments/concerns of Reviewers 2 and 3 and we have indicated any changes made in the manuscript to address these additional comments/concerns.

2) Optionally, the authors could consider revising the manuscript according to the second reviews listed below.

As noted above, we have also responded to the additional comments/concerns of Reviewers 2 and 3 in the sections that follow and, where appropriate, we have indicated any revisions made in the manuscript.

Reviewer #1 (Recommendations for the authors):The authors have addressed all the concerns. I have no further comments.

Thank you again for your constructive and helpful comments.

Reviewer #2 (Recommendations for the authors):The authors have responded at length in the "Response to Reviews" document, although in many cases, the actual changes in the manuscript are fairly minimal. In my opinion, several of the points that were raised in the initial review remain unclear or confusing in the presentation of the manuscript, as discussed below. It seems that several of these reflect an ambiguity about how (or even whether) measurements of "INaR" distinguish between resurgent sodium current and persistent sodium current.1) A point that was raised in in the previous reviews by both Reviewer 1 and Reviewer 2 is the interpretation of the experiment in Figure 6 (similar voltage-dependence of currents evoked by slow ramps either in the depolarizing or hyperpolarizing direction) as evidence that the voltage-dependence of "INaR" and "InaP" are the same. In response to the point raised by Reviewer 2 that the hyperpolarizing ramps may be almost all persistent sodium current because the resurgent current may be mostly inactivated before or during the hyperpolarizing ramp, the authors acknowledge this "excellent" point, but this isn't reflected in the actual manuscript. They have simply added lines 242-242 giving the original interpretation (that depolarizing ramps evoke only INaP and hyperpolarizing ramps INaP plus INaR), with no discussion of the size of the "extra" INaR relative to "INaP or the fact that "INaR" might be mostly inactivated. In fact, the current evoked by the hyperpolarizing ramp is the same magnitude as the depolarizing ramp, which seems most consistent with the idea that resurgent current has been totally inactivated before the hyperpolarizing ramps begins. In that case, ramps in both directions simply measure steady-state persistent current, and there is no justification for using this experimental result as evidence that the voltage-dependence of the resurgent current is the same as persistent current.

As noted above, we have added a section to the Methods section titled “Measurement of I_NaT_, I_NaR_, and I_NaP_” (lines 548-582). We have also added a statement to the Results section (where the experiment illustrated in Figure 6 is described) noting that I_NaR_ is inactivating during the hyperpolarizing ramp (lines 245-247).

2) A point that may be related: in both the original manuscript and the revised manuscript, it seems that often in referring to measurements of "INaR" – both for experiments and models – the authors are not distinguishing between components of resurgent current and persistent current, and the measurement of "INaR" refers to the combination of both. This is never really made clear in the manuscript, either in Methods or Results. The fact that they may consider "INaR" to refer both to decaying and steady-state components of current is suggested by the author's response to the issue of the time constant for decay of resurgent current predicted by the Raman-Bean 2001 model. The authors argue that their value of a time constant of 45 ms for the decay of resurgent current at -30 mV predicted by this model was not a mistake, but rather some sort of error in the 2001 paper in reporting a value of 25 ms for the Raman-Bean model ("We are not certain how the value of 25.4 ms reported in Raman and Bean, 2001 was determined."). In fact, it seems very clear that the difference is that the exponential fit in the Raman-Bean paper is only the decaying part of the current, i.e. decay of the resurgent current to a non-zero steady-state value that reflects INaP. This is clearly stated in that paper, and is clearly illustrated in the fits shown for both the experimental data (Figure 1B) and the results of the model (Figure 8B). To check if the authors' value of 45 ms could reflect some discrepancy in the actual modeling here and in the 20021 Raman-Bean paper, I digitized the trace the authors include in the Response to Reviews showing their calculation for resurgent current at -30 mV predicted by Raman-Bean model. With crude digitization of the trace, the trace is fit by a time constant is 27 ms, not 45 ms – if the fit allows for the steady-state value of non-decaying current. The authors do not show a fit to their trace, and the trace is truncated before the steady-state is clearly shown, but there is no way the trace is well-fit by an exponential with a time constant of 45 ms. The only way I can see of coming up with a time-constant of 45 msec would be to force a fit that decays to zero, which is clearly inappropriate for a current that decays to a steady-state.

Thank you for raising this point again. After reading your comments, we reexamined the data plotted in Figure 7I, and determined that we had made an error by neglecting to subtract I_NaP_, affecting our measurements of rate of I_NaR_ decay. We have corrected this error, subtracting I_NaP_ prior to calculating the decay taus. We have revised Figure 7I to reflect the corrected analyses. Thank you again.

These are details that do not detract from the main point of the manuscript that resurgent current can be interpreted without invoking an open-channel blocking mechanism. However, readers will benefit by a clear statement about whether and how components of NaR and INaP are distinguished in the various measurements that are reported, both for the experimental data and for both models that are discussed.

Thank you again. As noted above, we have now added a section to the Methods (lines 548-582) to clarify how we measured I_NaR_ and I_NaP_ in the experimental and modeling studies.

Reviewer #3 (Recommendations for the authors):The authors addressed all the questions raised by the reviewers in considerable detail. However, they did not touch the model nor did they revise their interpretation of the model. Therefore, my issues previously raised remain at the core. Below I have tried to clarify my concerns. Taken together the new model is not so different from the Raman and Bean (RandB) model. While it fares better in some aspects, it introduces new issues, especially with INaT inactivation and persistent current. Furthermore, it might be that the RandB model could approximate the new data far better than anticipated here, if only optimized (and maybe tested with a second IF =OB state). Finally, I do not agree with the interpretation that INaT and INaR are two completely distinct identities in their model as outlined below.INaT and INaR decay:The authors did not follow the implications of my comment. Because of the single open state INaT and INaR must! decay with the same kinetics. Apparently, they do as shown in Figure 2D at -45mV. The different decay is only due to using different voltages for comparison and of course different behavior of the IF/IS states at different voltages. Therefore INaT and INaR mechanisms are only distinct because of membrane potential (or its history) and not per se. The wording throughout the results part is still not reflecting this matter.

We think that adherence to a single time constant of decay would only be relevant if there were only a single conducting state and a single non-conducting state (two kinetic states total). With more than one non-conducting state, such as in our model and the Raman-Bean model, multiple components of decay from a single open state are possible and often observed. In Raman and Bean, 2001, for example, bi-exponential decay of the Nav currents in Purkinje neurons was shown experimentally (see Figure 4), observations that are consistent with our experimental and modeling results here.

In addition, in Figure 4B during bulk INaT IS, IF1 and IF2 become all populated quiet fast indicating IS is not a slow inactivated state, just a bit slower at depolarized potentials compared to IF. If stated this way, the new model is not very different from the RandB model. My interpretation of the new model is that at high potentials IF states adsorb fast and dissipate slowly (Figure 4B) while at lower potentials the equilibrium changes and they dissipate fast and do not adsorb (Figure 4B). Note, this is not different from the OB state in the RandB model (despite two IF states here). Second, the IS state is always absorbing and becomes dominant at more negative potentials. Therefore the IS state is not qualitatively different from the inactivated states in the Raman and Bean model. Note how similar the occupancies are in Figure 4A+B (->IS) vs. Figure 7B (->I). However, the new model differs in two important aspects, a second IF state and a shortcut from IF to C. Are these additions required? See my comment below.

We would like to re-iterate that our goal was not to identify inadequacies in the Raman-Bean model. Also, as indicated in the manuscript, we did not construct our model using the Raman-Bean model as a template. Rather, we constructed our model using our experimental data as constraints, and we developed a Markov model with parallel inactivation pathways that fit our experimental data well. We are presenting these experimental and modeling results to show that it is possible to generate I_NaR_ in the absence of an open-channel blocking step/mechanism.

Fast inactivation:The many inactivation time constants result in a strange looking slow decay component, not compatible with real data. This is becoming obvious in Figure 3H,I green traces. That is a kink in INaT and a slow decrease of INaP presumably due to slow accumulation in IS. Pointing out that three time constants of inactivation (+ transition to C) are likely too much.The fit of real data shown for reference in the response of the authors is not convincing, also because all components would not be covered with a single-exponential fit.

It has previously been demonstrated that voltage-steps between -45 mV and -30 mV (in Purkinje neurons) result in Nav currents with bi-exponential decay kinetics, as illustrated in Raman and Bean, 1997 and 2001. Our experimental data and model are also consistent with bi-exponential decay of the Nav currents.

Kinetic scheme:I do not agree with the authors here on one point important for the proposed model. In the Raman-Bean model, the blocked state could principally slowly dissipate through an increased persistent current, which is indeed visible in Fig, 7H. If the new data (with or without adding a second IF state to OB) is fitted to the RandB model, is the modified model then capable of fitting the decreased INaR and the data from Figure 8 and therefore not inferior to the proposed model here?

It is certainly possible (indeed likely) that adding states to the Raman-Bean model would allow us to better fit the model to our experimental data. Also, as you mention, it is possible that Nav channels may dissipate from the OB state into a slow-inactivated state. As indicated previously, however, our intention was not to revise or correct the Raman-Bean model. Rather, our objective was to explore the possibility that another mechanism could account for the gating of I_NaR_. We think these efforts are highly relevant and will be of considerable interest given that the targeted deletion of key candidate blocking particles does not abolish I_NaR_ (see Ransdell et al., 2017, White et al., 2019).

Differential recovery of INaT and INaR:I absolutely agree that two populations of sodium channels could be the cause as pointed out in the discussion. Given that, the arguments brought forward (line 214-233) are becoming void.

Thank you for this comment. We have added an additional comment (lines 218-219) regarding our hypothesis and motivation for the experiments presented in Figure 5.